# Relationship between HLA genetic variations, COVID-19 vaccine antibody response, and risk of breakthrough outcomes

Junqing Xie [1], Beatriz Mothe [2], Marta Alcalde Herraiz[1], Chunxiao Li[3], Yu Xu [4,5,6], Annika M. Jödicke[1], Yaqing Gao[7], Yunhe Wang[7], Shuo Feng [8], Jia Wei [9], Zhuoyao Chen [10], Shenda Hong [11,12], Yeda Wu [13], Binbin Su[14], Xiaoying Zheng[14,18], Catherine Cohet [15,18], Raghib Ali[3,16,18], Nick Wareham [3,18] & Daniel Prieto Alhambra [1,17] ✉

The rapid global distribution of COVID-19 vaccines, with over a billion doses administered, has been unprecedented. However, in comparison to most identified clinical determinants, the implications of individual genetic factors on antibody responses post-COVID-19 vaccination for breakthrough outcomes remain elusive. Here, we conducted a population-based study including 357,806 vaccinated participants with high-resolution HLA genotyping data, and a subset of 175,000 with antibody serology test results. We confirmed prior findings that single nucleotide polymorphisms associated with antibody response are predominantly located in the Major Histocompatibility Complex region, with the expansive HLA-DQB1*06 gene alleles linked to improved antibody responses. However, our results did not support the claim that this mutation alone can significantly reduce COVID-19 risk in the general population. In addition, we discovered and validated six HLA alleles (A*03:01, C*16:01, DQA1*01:02, DQA1*01:01, DRB3*01:01, and DPB1*10:01) that independently influence antibody responses and demonstrated a combined effect across HLA genes on the risk of breakthrough COVID-19 outcomes. Lastly, we estimated that COVID-19 vaccine-induced antibody positivity provides approximately 20% protection against infection and 50% protection against severity. These findings have immediate implications for functional studies on HLA molecules and can inform future personalised vaccination strategies.

The COVID-19 pandemic has led to an unprecedented development, approval, and rollout of vaccines in the history of vaccinology. As of June 8, 2023, over 13 billion doses have been delivered worldwide, with over 70% of the global population receiving at least one dose[1]. The "one-size-fits-all" vaccination strategy[2,3] has shown remarkable variability in impact across different subpopulations[4–6]. For the COVID-19 vaccine, this heterogeneity has led to a considerable number of fully vaccinated individuals remaining susceptible to COVID-19, requiring further booster doses.

Historically, immunogenetic variations, particularly the human leucocyte antigens (HLA) genes, have been recognised as

significant influencers of adaptive immune responses to various vaccines, including hepatitis B, measles, and influenza[7–9]. However, the role of HLA in the context of COVID-19 vaccines, a novel exogenous antigen, remains largely elusive[10]. A recent genome-wide association study (GWAS)[11] focusing on post-vaccination antibodies against COVID-19 unveiled a number of genome-wide significant single nucleotide polymorphisms (SNPs) and pinpointed HLA-DQB1:06, DQB1:06:02 subtype in particular, as the potential causal alleles. Because of the complex structure in the major histocompatibility complex (MHC) region, including intense gene density, high polymorphism, and long linkage disequilibrium, the statistically fine-mapping of GWAS output proximally near this region is subject to critical methodological challenges and limitations[12,13]. As a result, the predicted DQB1:06 alleles likely represent a fraction of HLA allelic variations that influence heterogeneous antibody response to COVID-19 vaccines, offering a glimpse into the larger, more comprehensive picture of its genetic basis. Furthermore, in the same study, HLA-DQB1:06 alleles were found to confer over 30% lower risk of breakthrough infection among a cohort of trial participants. The extent to which this association can be generalised to a broader general population is however unclear[14].

This population-based research consisted of three interconnected analyses to fill existing knowledge gaps. Firstly, we aimed to replicate prior findings on the effect of HLA-DQB1:06 alleles on enhancing vaccine-induced antibody responses and reducing SARS-COV-2 breakthrough infection risk. Secondly, we aimed to discover and validate novel genetic associations between HLA alleles and antibody response, and to investigate their clinical significance. Finally, we used Mendelian randomisation analyses to estimate the portion of effectiveness against SARS-CoV-2 infection and against severe COVID-19 attributable to vaccine-induced antibodies.

## Results

Figure 1 summarises the impact of eligibility criteria for participants of each analysis. Out of the 194,371 individuals who enroled in the SARS-CoV-2 antibody and infection seroprevalence study, 175,000 who self-reported having received either one or two doses of COVID-19 vaccines at the time of antibody testing and had no previous COVID-19 infection as confirmed by the absence of antibodies against SARS-CoV-2 nucleocapsid antigen were included for the cross-sectional (CS) cohort. Baseline characteristics of the overall, discovery, and validation CS cohorts are detailed in Table 1. Participants were stratified according to whether they had received one dose (CS-1-dose) or two doses of the COVID-19 vaccine (CS-2-dose), and further analyses were conducted accordingly. Following a median interval of 52 days (interquartile range: 36–64 days) after the first dose of vaccination, 28.4% of recipients tested positive for anti-spike SARS-CoV-2 antibodies. This proportion escalated to 65.5% among individuals who had the antibody test a median of 20 days (interquartile range: 8–33 days) after administration of the second dose of the vaccine.

In the prospective cohort (PS) of 357,806 individuals with recorded COVID-19 vaccination in the linked primary care database, the mean age was 69.3 years (SD: 8.0) with 55.3% being female (baseline demographics shown in Supplementary Table 1). During the one-year follow-up period post-first dose vaccination, a total of 17,068 individuals experienced a breakthrough infection as confirmed by polymerase chain reaction test, and 1,353 were hospitalised with or died from COVID-19.

### Verification and extension of previous key findings

In the GWAS of the first-dose antibody positivity, the lead SNP, rs2150392827 (position: 6:32451297; OR, 0.865; $P$, $1.27 \times 10^{-25}$), was located in the chromosome 6 MHC region with HLA genes to be nearest one (Fig. 2A). Similar GWAS signals were found for the second-dose antibody positivity, with the lead SNP rs11490315 (position:

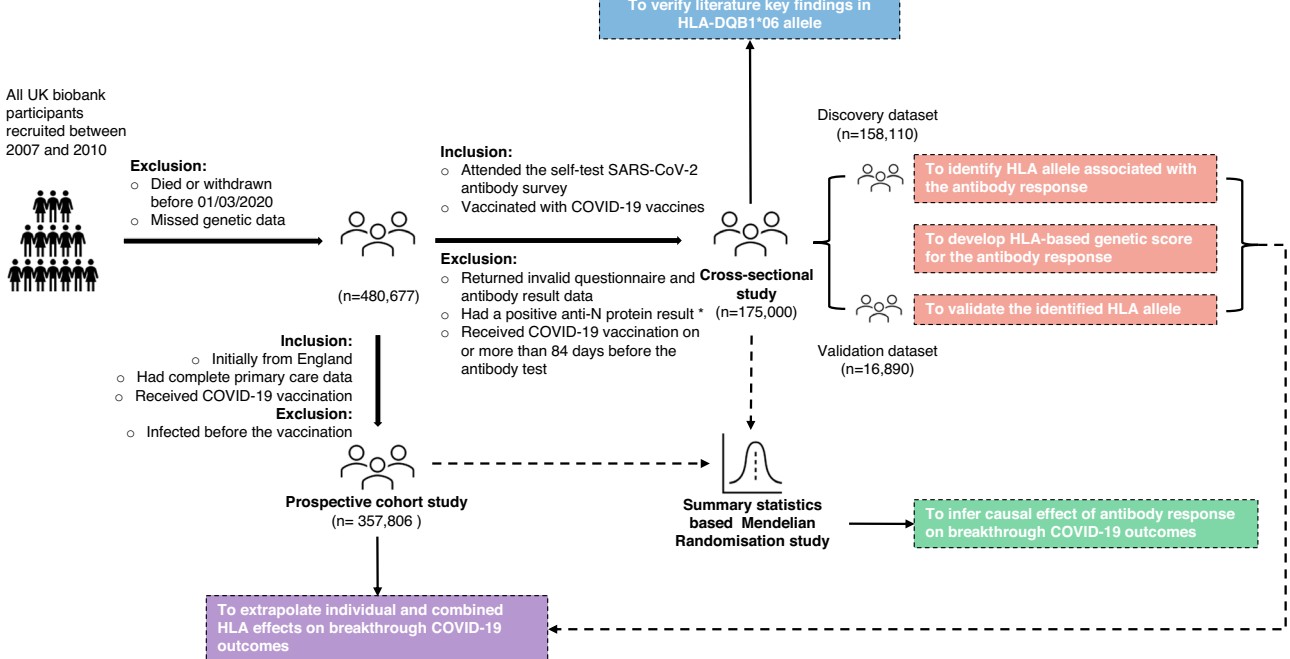

**Fig. 1 | Study design and flowchart.** Information about previous COVID-19 infection status was not available for all participants. Specifically, out of the 81,353 people who tested positive in the self-test antibody study, only 58,821 (72.3%) participated in the follow-up infection study and had a valid anti-N protein result. Of these, 10,783 (18.3%) tested positive for the anti-N protein and were excluded from the study. For the remaining 22,532 (27.6%) individuals who did not participate in the follow-up infection study, and thus had an undetermined prior infection status, we made an assumption that they were COVID-19 naïve and were included in the subsequent analyses. As a result, it is possible that about 5.0% (27.6% × 18.3%) of antibody-positive participants in the current cohort could result from the previous infection rather than the vaccination.

**Table 1 | Baseline characteristics of eligible participants with the antibody test in the cross-sectional cohort**

| | First-dose vaccine recipients | | | Second-dose vaccine recipients | | |
|---|---|---|---|---|---|---|
| | All | Discovery dataset | Validation dataset | All | Discovery dataset | Validation dataset |
| Number | 107,175 | 97,097 | 10,078 | 67,825 | 61,013 | 6812 |
| Age, mean (SD) | 67.27 (7.59) | 67.23 (7.60) | 67.61 (7.53) | 71.22 (6.56) | 71.21 (6.57) | 71.29 (6.50) |
| Age category, n (%) | | | | | | |
| 50–59 years | 42,872 (40.0) | 38,671 (39.8) | 4201 (41.7) | 45,931 (67.7) | 41,302 (67.7) | 4629 (68.0) |
| 60–69 years | 19,970 (18.6) | 18,223 (18.8) | 1747 (17.3) | 5085 (7.5) | 4582 (7.5) | 503 (7.4) |
| ≥70 years | 44,333 (41.4) | 40,203 (41.4) | 4130 (41.0) | 16,809 (24.8) | 15,129 (24.8) | 1680 (24.7) |
| Sex, n (%) | | | | | | |
| Female | 59,761 (55.8) | 54,584 (56.2) | 5177 (51.4) | 39,417 (58.1) | 35,733 (58.6) | 3684 (54.1) |
| Male | 47,414 (44.2) | 42,513 (43.8) | 4901 (48.6) | 28,408 (41.9) | 25,280 (41.4) | 3128 (45.9) |
| Ethnicity | | | | | | |
| White | 90,511 (84.5) | 81,825 (84.3) | 8686 (86.2) | 56,814 (83.8) | 50,933 (83.5) | 5881 (86.3) |
| Others | 16,664 (15.5) | 15,272 (15.7) | 1392 (13.8) | 11,011 (16.2) | 10,080 (16.5) | 931 (13.7) |
| Weeks since the latest vaccination, n (%) | | | | | | |
| 1 week | 1677 (1.6) | 1513 (1.6) | 164 (1.6) | 13,222 (19.5) | 11,941 (19.6) | 1281 (18.8) |
| 2 weeks | 2957 (2.8) | 2683 (2.8) | 274 (2.7) | 12,911 (19.0) | 11,624 (19.1) | 1287 (18.9) |
| 3 weeks | 4609 (4.3) | 4173 (4.3) | 436 (4.3) | 10,715 (15.8) | 9675 (15.9) | 1040 (15.3) |
| 4 weeks | 7053 (6.6) | 6408 (6.6) | 645 (6.4) | 8939 (13.2) | 7963 (13.1) | 976 (14.3) |
| 5 weeks | 9188 (8.6) | 8361 (8.6) | 827 (8.2) | 7018 (10.3) | 6295 (10.3) | 723 (10.6) |
| 6 weeks | 10,489 (9.8) | 9558 (9.8) | 931 (9.2) | 4989 (7.4) | 4483 (7.3) | 506 (7.4) |
| 7 weeks | 12,350 (11.5) | 11,171 (11.5) | 1179 (11.7) | 3426 (5.1) | 3068 (5.0) | 358 (5.3) |
| 8 weeks | 14,840 (13.8) | 13,424 (13.8) | 1416 (14.1) | 2438 (3.6) | 2211 (3.6) | 227 (3.3) |
| 9 weeks | 15,613 (14.6) | 14,088 (14.5) | 1525 (15.1) | 1718 (2.5) | 1543 (2.5) | 175 (2.6) |
| 10 weeks | 14,655 (13.7) | 13,263 (13.7) | 1392 (13.8) | 1275 (1.9) | 1153 (1.9) | 122 (1.8) |
| 11 weeks | 11,319 (10.6) | 10,272 (10.6) | 1047 (10.4) | 733 (1.1) | 650 (1.1) | 83 (1.2) |
| 12 weeks | 2425 (2.3) | 2183 (2.2) | 242 (2.4) | 441 (0.7) | 407 (0.7) | 34 (0.5) |
| Vaccine types, n (%)[a] | | | | | | |
| ChAdOx1 | 39,948 (37.3) | 36,366 (37.5) | 3582 (35.5) | 18,578 (27.4) | 16,721 (27.4) | 1857 (27.3) |
| BNT162b2 | 15,188 (14.2) | 13,690 (14.1) | 1498 (14.9) | 17,587 (25.9) | 15,886 (26.0) | 1701 (25.0) |
| Unknown | 52,039 (48.6) | 47,041 (48.4) | 4998 (49.6) | 31,660 (46.7) | 28,406 (46.6) | 3254 (47.8) |

[a]Calculated based on a subgroup of participants who had complete linkage to primary care records for information on COVID-19 vaccine types.

6:32663047; OR, 0.863; $P$, $3.89 \times 10^{-11}$), also nearest to HLA genes (Fig. 2E).

In the candidate HLA-gene-based analysis, 42.1% of people in the overall CS cohort carried at least one HLA-DQB1:06 allele, a proportion similar to the previous report of 44.1% in the trial's participants. The distribution of temporal duration between the date of vaccination and the date of antibody testing was similar between carriers and non-carriers of the HLA-DQB1:06 alleles, which verified our assumption that genetic variations were independent of this critical determinant for the antibody positivity (Supplementary Table 2). On average, the antibody response post-first dose was positive in 29.5% of HLA-DQB1:06 alleles carriers, compared to 27.4% in non-carriers. This difference in antibody positivity remained consistent over time and was observed up to eight weeks after the initial vaccination (Fig. 2B). A higher but attenuated rate of antibody positivity was observed post-second dose in people carrying the DQB1*06 alleles (66.1%) than those without carrying the DQB1*06 alleles (65.2%) (Fig. 2F).

After adjusting for potential confounding factors, a significant positive association was found between HLA-DQB1*06 alleles and the antibody response in the CS-1-dose cohort (odds ratio OR 1.13, 95% CI 1.10 to 1.17) and in the CS-2-dose cohort (OR 1.04, 95% CI 1.00 to 1.07). When considering the five specific subtypes of DQB1:06 alleles (Fig. 2C), the DQB1:06:04 subtype showed a markedly stronger association with vaccine-induced antibodies, compared to the DQB1:06:02 subtype, which was previously hypothesised to be primarily responsible for binding to the SARS-CoV-2 spike protein more effectively. Furthermore, the frequency of DQB1:06 alleles showed no association with COVID-19 outcomes after vaccination (therefore breakthrough COVID-19) (Fig. 2D), with adjusted OR corresponding to 0.98 (95% CI 0.94 to 1.02) for infection, and 1.00 (95% CI 0.88 to 1.13) for severity, defined as hospitalisation with or death from COVID-19.

**Discovery and validation of novel HLA allele associations with antibody response**

We then proceeded to identify novel HLA associations with antibody response by conducting a gene-based analysis across 213 classic alleles in 11 HLA genes. In general, the associations were more pronounced in relation to the first-dose (Fig. 3A left) than the second-dose (Fig. 3A right) antibody response.

In total, 15 alleles of eight HLA genes (A, C, DQA1, DQB1, DRB1, DRB3, DRB5, and DPB1) retained statistical significance in both the discovery and validation dataset after FDR-correction (Supplementary Table 3). Among them, 13 HLA alleles were associated with first-dose antibody response, 1 with second-dose antibody response, and 1 with both (Fig. 3B). Nine HLA alleles were linked to a higher antibody positivity (enhancer), while six alleles were linked to lower antibody positivity (suppressor).

The HLA-DQB1*06:04 allele was the most potent enhancer of antibody response ($\beta = 0.213$ in discovery and 0.384 in validation)

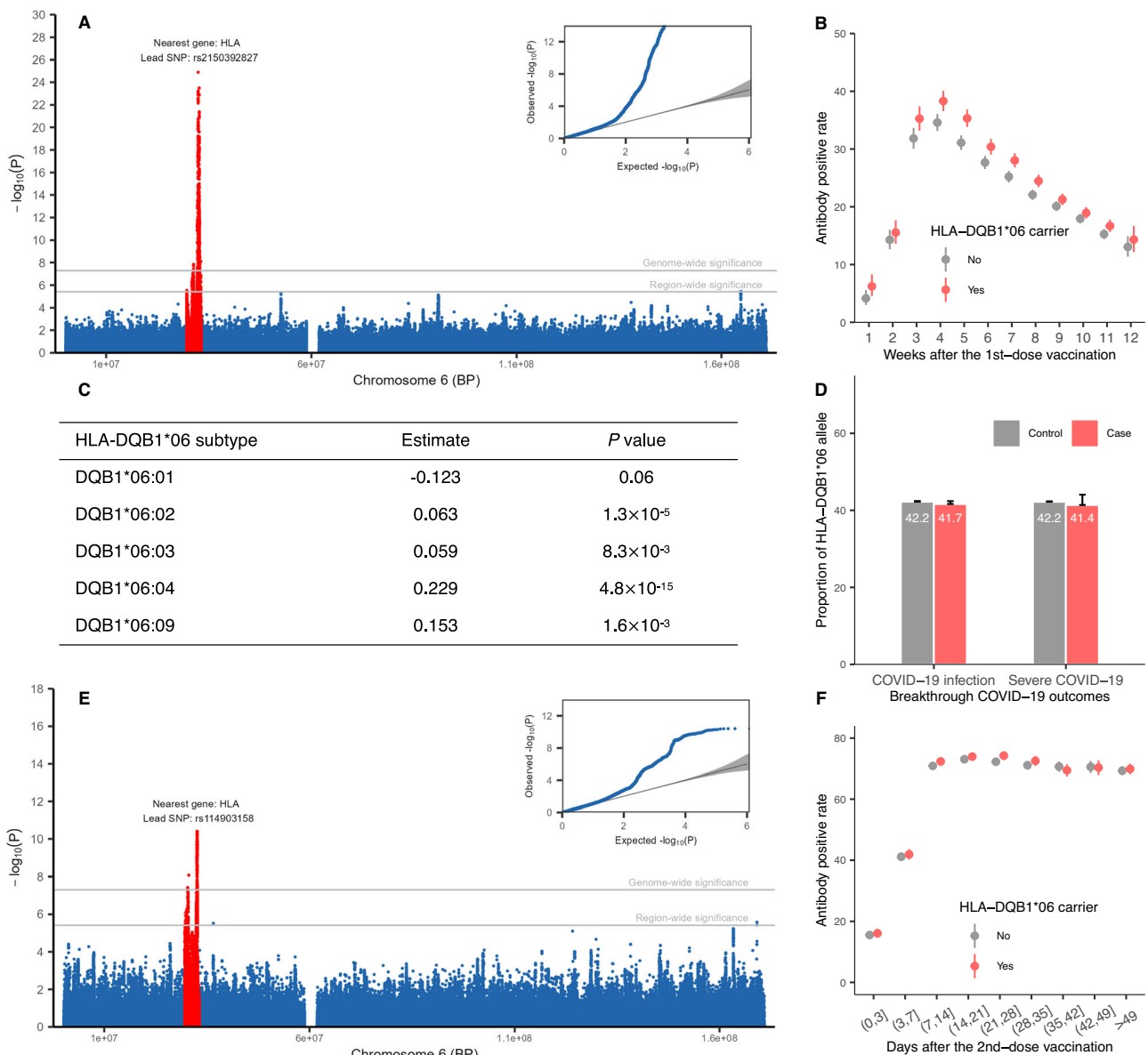

**Fig. 2 | Verification and extension of findings on HLA-DQB1\*06 alleles.**
**A** Manhattan plot of GWAS in chromosome 6 on antibody response to first-dose COVID-19 vaccine. SNPs within the MHC region are labelled in red colour, with the nearest gene to the lead SNP presented. The whole GWAS results are shown in Supplementary Fig. 1. **B** The proportion of individuals with a positive antibody response, stratified by the HLA-DQB1\*06 status in the CS-first-dose cohort. Individuals with one or two DQB1\*06 alleles are classified as carriers, whereas those with zero DQB1\*06 alleles are non-carriers. **C** The association of five specific HLA-DQB1\*06 subtypes with the antibody response. The estimate presents the coefficient of each allele in the logistic regression model. **D** The proportion of individuals carrying HLA-DQB1\*06 alleles, comparing COVID-19 cases with controls (those not

infected). Specific numerical data is labelled with white text within each bar.
**E** Manhattan plot of GWAS in chromosome 6 on antibody response to second-dose COVID-19 vaccine. SNPs within the MHC region are labelled in red colour, with the nearest gene to the lead SNP. The entire GWAS results are shown in Supplementary Fig. 1. **F** The proportion of individuals with a positive antibody response, stratified by the HLA-DQB1\*06 status in the CS-second-dose cohort. The error bars in panels **B**, **D**, and **F** indicate the two-sided 95% confidence interval without correcting for multiple testing. The *P* value of the Wald test in panels **A**, **C** and **E** is presented raw and did not correct for multiple testing. Source data are provided in the Source data file. The inset of the Manhattan plot on the left of the figure shows a Q–Q plot, with expected *P* values on the *x*-axis and observed *P* values on the *y*-axis.

following first-dose vaccination. However, no effect was detected for this allele after the second dose. In contrast, DQA1\*01:01 consistently functioned as an antibody suppressor either following the first ($\beta = -0.122$ in discovery and $-0.180$ in validation) or second vaccination ($\beta = -0.137$ in discovery and $-0.165$ in validation). Several haplotypes or clusters of HLA alleles were identified among all validated alleles (Fig. 3C): Cluster 1: DQA1\*01:02; DRB5\*01:01; DRB1\*15:01, Cluster 2: DRB3\*01:01; DRB1\*03:01; DQB1\*02:01, Cluster 3: DRB1\*01:01; DQB1\*05:01; DQA1\*01:01, Cluster 4: DQB1\*06:04; DRB3\*03:01; DRB1\*13:02.

After removing highly correlated alleles from each cluster, seven independent HLA alleles that are most significantly associated with antibody response were DQB1\*06:04, DQA1\*01:02, DRB3\*01:01, C\*16:01, DPB1\*10:01, A\*03:01, DQA1\*01:01. All alleles, except for A\*03:01, appeared to have an additive effect on antibody response (Fig. 3D) and presented consistent associations between White and other ethnic groups (Supplementary Fig. 2).

We constructed genetic scores (GS) of antibody response to assess its combined impact by aggregating all HLA allelic variations with different statistical thresholding. The distribution of each GS is

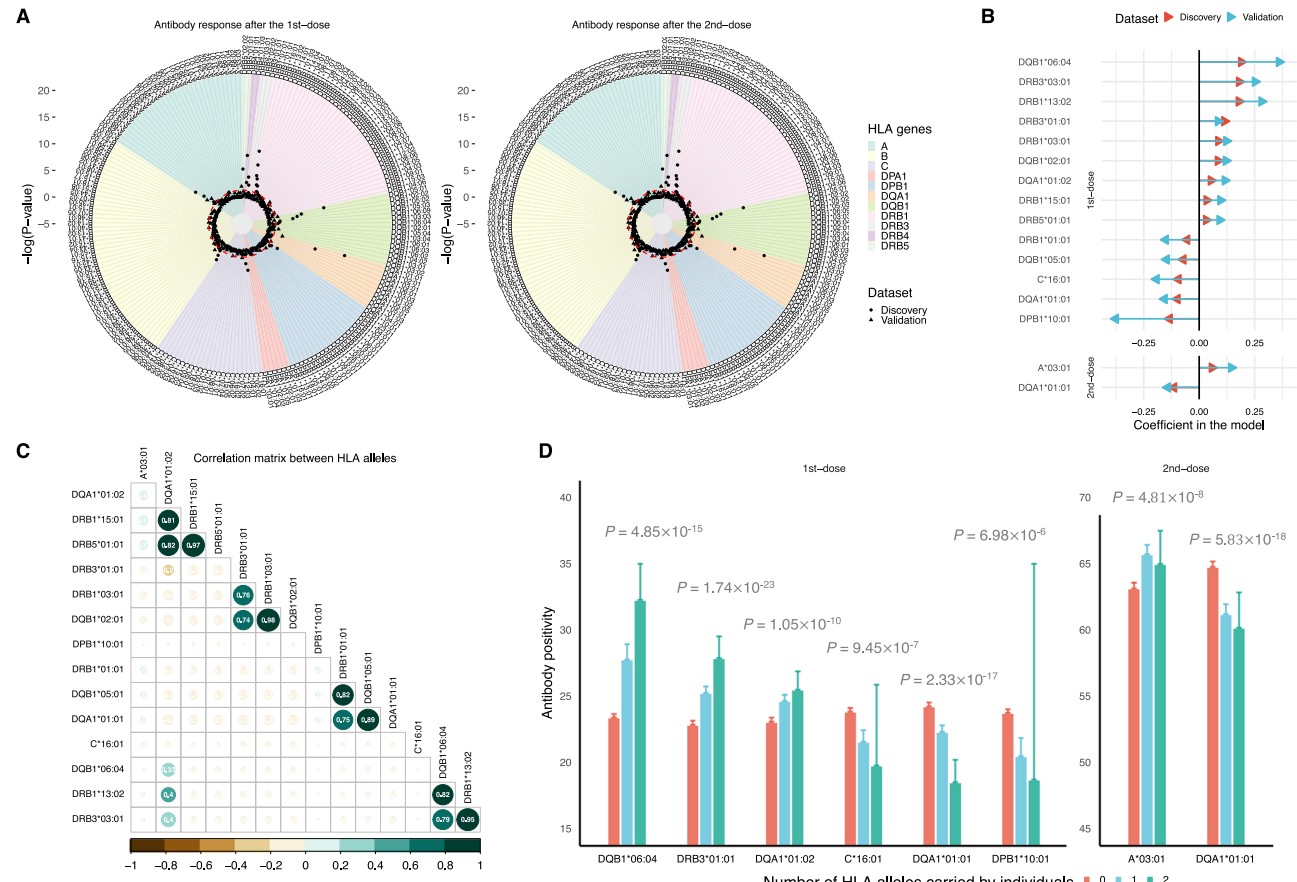

**Fig. 3 | Association of 203 candidate HLA alleles with COVID-19 vaccine antibody response. A** Circular Manhattan plot of HLA-gene-based genetic analysis for the first-dose antibody response (left panel) and second-dose antibody response (right panel) in the discovery dataset. The red dashed circle indicates a threshold of *P* value of 0.05. **B** Association strength and direction of 15 HLA alleles that are significant in both the discovery and validation datasets following the false discovery rate correction of the total number of alleles tested. The upper portion of the panel represents the results for the first-dose antibody response, while the lower portion represents the second-dose antibody response. **C** Pairwise correlation plot between 15 significant HLA alleles. **D** Antibody positivity rate among individuals with varying copies of alleles (0, 1, 2) across seven independent and significant HLA genes. The left portion of the panel represents the results for the first-dose antibody response, while the right portion is for the second-dose antibody response.

provided in Supplementary Fig. 3. The results showed a significant association of the GS with antibody response. More specifically, for GS1, an adjusted OR of 1.14 (95% CI 1.13 to 1.16) per SD increase was found for antibody positivity following first-dose vaccination, and an OR of 1.10 (95% CI 1.08 to 1.11) for 2-dose antibody positivity. For GS2, the corresponding ORs were 1.16 (95% CI 1.14 to 1.18) and 1.12 (95% CI 1.10 to 1.14), respectively. GS3 presented the strongest association (1.17 [95% CI 1.15 to 1.18] for the first dose, and 1.14 [95% CI 1.12 to 1.15] for the second dose) and was selected for subsequent analyses (Supplementary Table 4).

**Extrapolation of HLA effects on COVID-19 breakthrough outcomes**

Among the seven validated HLA alleles with associations with either first-dose or second-dose antibody response, none of them was individually associated with clinical outcomes of breakthrough COVID-19 (Supplementary Table 5). However, a clear dose-dependent correlation between antibody response and COVID-19 outcome predisposed by each HLA allele was observed (Fig. 4A, B). Individuals who carried an allele that enhanced antibody response had a numerically (but not significant) lower risk of breakthrough COVID-19 outcomes, with HLA-DQB1:06:04 showing the strongest effects on both breakthrough infection (Hazard ratio HR 0.94 95% 0.88 to 1.01) and severe COVID-19 (HR 0.90 95% 0.70 to 1.15). In general, severe COVID-19 was more

correlated with antibody response conferred by each HLA allele than the breakthrough infection outcome, as suggested by a steeper slope Fig. 4A, B.

In comparison with the GS of first-dose antibody response, GS of second-dose antibody response showed a stronger association with breakthrough COVID-19 outcomes, including infection (HR per GS unit increase 0.94, 95% CI 0.90 to 0.98) and hospitalisation or death (HR 0.87, 0.76 to 0.99). Furthermore, we identified 5% of participants who had a 26% (HR 1.26 95% CI 1.01 to 1.58) or a 34% (HR 1.34 95% CI 1.08 to 1.67) increased risk of severe COVID-19 based on first-dose or second-dose GS, respectively (Fig. 4C, D and Supplementary Table 5).

Finally, we estimated the causal effect of vaccine-induced antibody positivity on breakthrough COVID-19 outcomes. Based on the seven validated independent alleles used as instrumental variables in Mendelian randomisation analyses, we demonstrated effectiveness of antibody positivity ranging from 16.84% (95% CI 6.39% to 26.12%) to 21.68% (5.99% to 34.76%) in preventing breakthrough infection, and 46.44% (95% CI −1.49% to 71.73%) to 49.94% (95% CI 15.52% to 70.34%) in preventing severe COVID-19 (Table 2). Sensitivity analyses produced consistent results (Supplementary Table 6).

## Discussion
The present study is the largest and most comprehensive to investigate the genetic determinants of antibody response to COVID-19

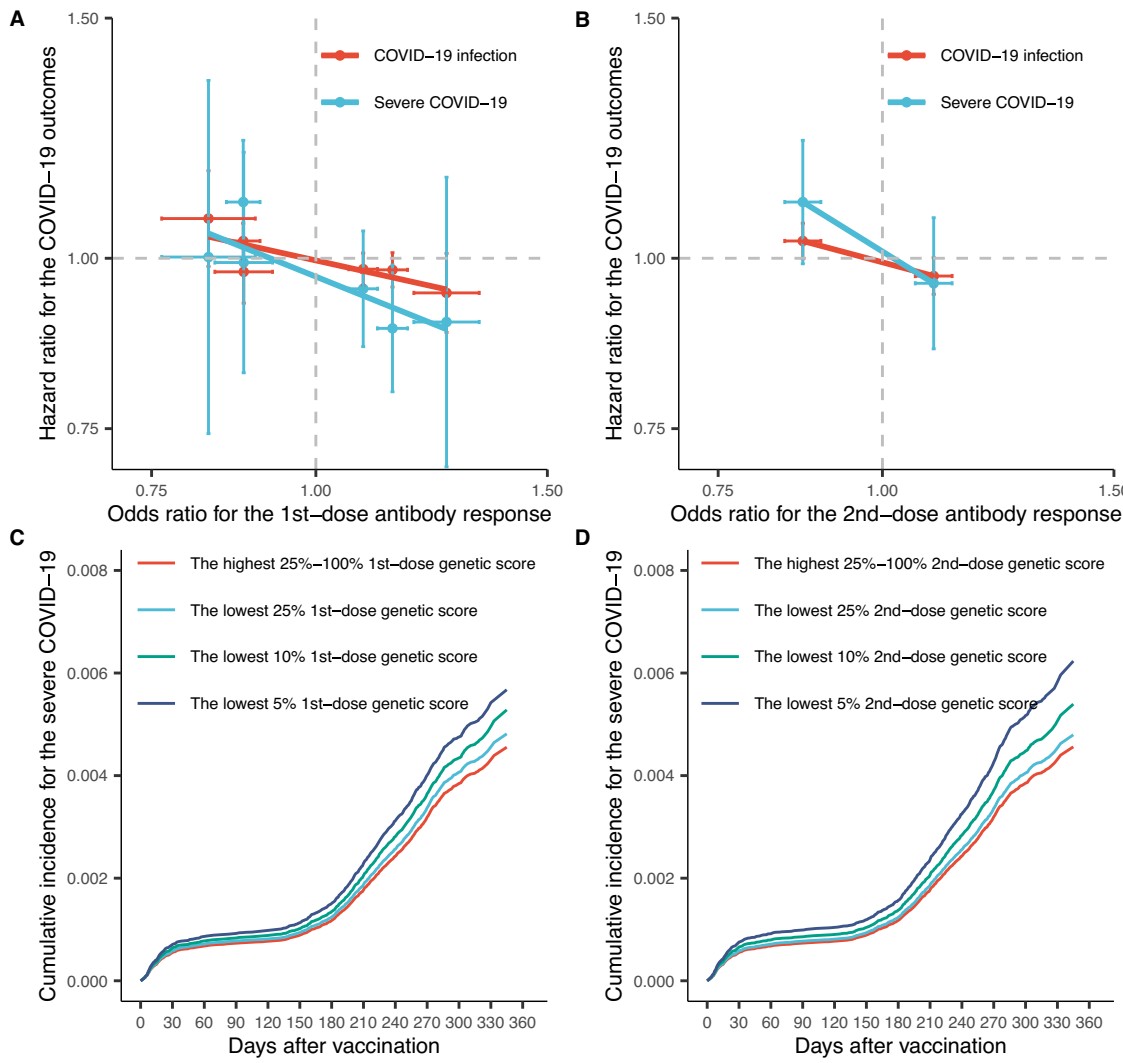

**Fig. 4 | Extrapolation of individual and combined HLA effects on breakthrough COVID-19 outcomes. A** Correlation between the effect of six independent HLA alleles on first-dose antibody response and their subsequent effects on breakthrough COVID-19 outcomes. **B** Correlation between the effect of two independent HLA alleles on second-dose antibody response and their subsequent effects on breakthrough COVID-19 outcomes. Age–sex-adjusted cumulative incidence curve of severe COVID-19 stratified by the first-dose (**C**) and second-dose (**D**) genetic scores.

**Table 2 | Effect of vaccine-induced antibody response on breakthrough COVID-19 outcomes**

| Exposure | Breakthrough outcomes | Number of HLA alleles | Relative risk (95% CI) | Effectiveness (95% CI) |
|---|---|---|---|---|
| First-dose antibody response | COVID-19 infection | 6 | 0.83 (0.74 to 0.94) | 16.84 (6.39 to 26.12) |
|  | Severe COVID-19 | 6 | 0.55 (0.36 to 0.84) | 49.94 (15.52 to 70.34) |
| Second-dose antibody response | COVID-19 infection | 2 | 0.78 (0.65 to 0.94) | 21.68 (5.99 to 34.76) |
|  | Severe COVID-19 | 2 | 0.54 (0.28 to 1.01) | 46.44 (−1.49 to 71.73) |

vaccines and their impact on subsequent clinical outcomes, with a particular focus on the HLA genes. Firstly, we confirmed earlier findings that the HLA-DBQ1*06 allele family enhances the antibody response to initial COVID-19 vaccination. However, we refuted the notion that it can meaningfully impact clinical outcomes of breakthrough COVID-19 in isolation. Additionally, we found that the specific HLA-DBQ1*06:04 allele is likely the driving subtype, as opposed to the previously hypothesised HLA-DBQ1*06:02.

Secondly, we identified and validated six new HLA alleles that independently influence antibody response after COVID-19 vaccines. The cumulative effect of multiple HLA variants, represented in the form of a genetic score, was associated with both the susceptibility to, and the severity of breakthrough COVID-19.

Thirdly, we estimated that vaccine-induced antibody response alone confers approximately 20% protection against SARS-CoV-2 infection, and to a much greater extent (around 50%) against severe COVID-19 outcomes.

A few small candidate genetic studies have been undertaken to understand the relationship between HLA alleles and antibody responses to COVID-19 vaccines, with only two GWAS currently peer-reviewed[11,15]. The majority of these have been constrained by limited sample size, typically including dozens of individuals recruited from convenient populations, such as healthcare workers or clinical trial participants. The first significant HLA allele was reported by a Spanish study of 87 healthcare workers[16], showing a positive association between HLA-DRB1*07:01 and antibody levels 30 days post-receipt of

the second dose of the mRNA-1273 vaccine. This finding was partially supported by our data, with an elevated antibody positivity post-second dose vaccination amongst HLA-DRB1*07:01 carriers. However, this association did not meet statistical significance. Other studies involving healthcare workers from Japan (100 participants)[17], Italy (56 participants)[18], and the UK (251 participants)[19] have all failed to detect any significant associations between HLA alleles and anti-spike antibody response or seroprevalence following vaccination, except for the last UK study supporting DRB1*15:01 linked to stronger spike T-cell responses. In our study, DRB1*15:01 is one of the validated alleles associated with increased antibody rate. Evidence regarding the alleles B*40:02, DPB*106:01, A*24:02, and C*07:01, which were recently reported in association with antibody response, is conflicting[20]. Unfortunately, these associations were not confirmed, even though some of these alleles are highly correlated with DRB1*03:01, an allele that is significant in our research. Reassuringly, the additional DQB1*06:04 and A*03:01 alleles identified by the prior two GWAS were all independently verified by our study. Most recently, in a study that used the same source data as ours, the authors reported four independent alleles: DRB1*13:02, DQA1*01:01, DPB1*04:01, and DQB1*02:01, associated with the antibody response to the initial dose. All these significant alleles, except for DPB1*04:01, were identified in the present study, despite notable differences in participant eligibility criteria, cohort definition, antibody positivity determination windows, and analytic methods. Nevertheless, our study pinpoints DQB1*06:04 and DRB3*01:01 as the most likely causal HLA alleles, which are distinct from but in high linkage disequilibrium with DRB1*13:02 and DQB1*02:01, respectively, as found in their study. Further functional experiments are needed to corroborate or refute these statistical fine-mapping findings.

In addition to vaccine antibody response, a GWAS among 17,440 vaccinated participants reported a robust association between HLA-A*03:01 and vaccine-related side effects, such as chills and fever[21]. Interestingly, the HLA-A*03:01 allele was also associated with an enhanced antibody response in our study and the study of Italian healthcare workers vaccinated with BNT162b2 vaccine[15]. This observation may reflect phenotypic correlations between reactogenicity and immunogenicity traits following COVID-19 vaccination[22]. Whether there is a genetic colocalization mechanism involved between stronger antibody responses and higher side effects after vaccination via the HLA-A03:01 allele warrants further research. One potential biological hypothesis could be the cross-reactive highly restricted HLA-A*03:01 SARS-CoV-specific CD8 T cells, which existed prior to SARS-CoV-2 exposure to facilitate a higher response upon vaccination, as well as side effects.

To date, there is a paucity of data on the impact of HLA variations on the risk of post-vaccination breakthrough COVID-19 outcomes. One previous study based on pivotal trial data found a substantial reduction (over 30%) in breakthrough infection among HLA-DQB1*06 allele carriers[11]. However, we were unable to replicate this finding in our larger and more extensive community-based population-based cohort. This divergence could be attributed to immunosenescence[23,24], considering the relatively younger (mean age of 37 years) and healthier population in the prior study, and the design, as trial participants exclusively received the ChAdOx1 vaccine.

There is a clinical agreement that vaccination is efficacious and offers superior protection against severe COVID-19 than infection. This consensus is based on evidence gathered from initial vaccine trials, as well as routinely collected health data[25,26]. Our research is the first application of Mendelian randomisation to estimate COVID-19 vaccine effectiveness. This additional layer of genetic evidence supporting the protective role of vaccine-induced antibodies can further improve confidence in COVID-19 vaccines and tackle vaccine hesitancy. Moreover, our estimations provide quantitative evidence of the proportion of protection attributable to antibody-mediated

immunity following vaccination, informing the additional protection conferred by cellular immunity. Interestingly, a recent meta-analysis of 15 studies modelled that the efficacy of detectable neutralising antibodies may be approximately 50% against infection and 80% against severe COVID-19[27]. The numerical figures of our estimated antibody protection are lower than the prior extrapolated ones, but it is critical to acknowledge key differences between studies. Firstly, neutralising antibodies constitute only a part of the overall antibody repertoire. Secondly, the COVID-19 vaccines' efficacy and effectiveness may numerically differ in nature. Thirdly, the scale of Mendelian randomisation estimates for exposure represents the odds ratio per one unit increase in the log odds of genetic liability to seroconversion, which is distinct from the nominal seropositivity (yes vs no).

While current vaccines have shown substantial effectiveness, cases of non-responsiveness and breakthrough infections continue to persist, even following booster doses. Our study has produced a comprehensive map of HLA associations with vaccine-induced antibody responses. These most significant alleles should be prioritised in mechanistic research to elucidate their functional implications, thereby informing the development of novel vaccines. Compared to previous genetic risk scores that predicted COVID-19 outcomes by aggregating numerous SNPs from COVID-19 GWAS[28], our newly developed HLA genetic score is biologically understandable. This GS could be implemented through a tailored HLA-based genotyping array.

Although our investigation, focused on gene-level alleles, may be better equipped to identify functional HLA variants that are possibly missed in prior SNP-based GWAS, experimental validation remains essential for the associations identified. The antibody response process is dynamic and complex. Comparatively, the binary phenotype of antibody response lacks the statistical efficiency of continuous metrics, like antibody titre. Our study's size, over two hundred times larger than the next largest study on this subject, partially compensates for this drawback. Furthermore, antibody positivity offers limited insight into other immune characteristics, such as peak titer, durability, specificity, and neutralisation capacity of antibodies, and does not directly indicate their potential protective efficacy. A minor group of individuals showing a positive response could be attributed to COVID-19 infection instead of vaccination. This outcome misclassification, albeit not differing across genotypes, might result in the underestimation of the genetic influence on antibody response. Also, we ascertained COVID-19 status through routinely collected data, which is likely to capture more symptomatic and severe cases. Lastly, the genetic findings were primarily derived from European ancestry populations and caution should be taken when extrapolating to wider ethnic groups.

Using an extensive dataset incorporating host genetics, immunological biomarkers, vaccination statuses, and disease outcomes, alongside advanced methodologies, and multidisciplinary expertise within our team, we successfully identified six previously unreported HLA alleles that regulate the COVID-19 vaccine antibody response. Furthermore, we demonstrated that variations within the HLA strongly, collectively, and causally impact critical COVID-19 outcomes within a vaccinated population. These findings can inform vaccine developments by substantially advancing the understanding of genetic mechanisms of vaccine immunogenicity and clinical effectiveness, which may open new paths for personalised vaccination in the foreseeable future.

## Methods

### Study design and participants

**UK Biobank study.** The UK Biobank (UKBB) is a longitudinal population-based cohort of over 500,000 individuals aged 40–69 years at the time of recruitment between 2006 and 2010, recruited from England, Scotland, and Wales of the United Kingdom. Its study design and participant characteristics have been previously described

in detail elsewhere[29]. This study has received approval from the North West Multi-centre Research Ethics Committee under project application 65397. All participants gave their written informed consent. Further information can be found at www.ukbiobank.ac.uk/ethics.

On enrolment, participants responded to socio-demographic, lifestyle, and health-related questionnaires, undertook physical assessments, and provided electronically signed consent. Biological samples of blood, urine, and saliva were collected and stored to facilitate the use of diverse assays, such as genotyping data, as used in this investigation. Participants were granted permission to follow up on their health outcomes over an extended period through linkage to electronic health records, including data related to the COVID-19 pandemic.

**Self-test SARS-CoV-2 coronavirus antibody study.** Alive UKBB participants were re-invited for a SARS-CoV-2 coronavirus antibody study from February 2021 to July 2021, when the UK's vaccination programme was being rapidly implemented. At the initial study design stage, all potential participants of both sexes and age groups who met the predetermined inclusion and exclusion criteria were eligible for recruitment. These individuals were invited to participate through an email containing a brief overview of the study, links to the information sheet, an instructional video, a list of frequently asked questions, and the online consent form. A concerted effort was made to ensure that as many people as possible participated in this study. For instance, interested individuals were instructed to confirm their contact information and consent to receive a lateral flow self-test kit at their residence. Non-participants were also encouraged to confirm their non-participation via the same website. Upon consenting, participants received an acknowledgement email confirming their participation and providing details on when to expect their antibody testing kit. Participants were notified by email 1–3 days before kit shipment, and their addresses were securely transferred to a third-party mailing house and shipping company. A reminder email was sent to participants who had not returned a test result one week after their kit was dispatched.

Participant recruitment was done in two phases. The first phase targeted individuals who had previously attended a UKBB imaging assessment centre, with approximately ~34,713, ~22,390, and ~21,405 participants sequentially invited. The second phase invited the remaining ~371,985 participants who were not eligible for inclusion in phase 1 (more details on the study design, participant inclusion and exclusion criteria are provided in the online document: https://biobank.ndph.ox.ac.uk/showcase/label.cgi?id=998).

**SARS-CoV-2 coronavirus infection study.** As the lateral flow test (LFT) device used in the SARS-CoV-2 antibody study could not distinguish between antibodies produced in response to infection and those generated by vaccination, a follow-on COVID-19 infection study was conducted. Individuals who had previously participated in the self-test antibody study (phase 1 or phase 2) and had reported a positive antibody test result were re-invited to provide a capillary blood sample for laboratory analysis of specific antibodies that are solely produced in response to COVID-19 infection using the Elecsys® Anti-SARS-CoV-2 immunoassay. The assay uses a recombinant protein representing the nucleocapsid antigen in a double-antigen sandwich assay format, which favours the detection of high-affinity antibodies against SARS-CoV-2. The recruitment of participants for the COVID-19 infection study was similar to that of the self-test antibody study.

**Secondary data linkage**

Multiple electronic health records databases have been linked at the individual level to enable following up on the health and disease status of UKBB participants. The pertinent databases used in this study included primary care records (prescriptions and disease diagnoses),

hospital inpatient admission records (disease diagnoses), cause-specific death registers, and national infectious diseases surveillance data (COVID-19 results based on PRC)[30]. Our research team and other investigators have comprehensively scrutinised data linkage strategies across the different sources, and ensured that the quality and validity are suitable for research purpose[31,32].

**Analytic cohorts**

We generated a CS and PS cohort for current analyses. The CS cohort was composed of participants from the SARS-CoV-2 antibody and infection study, which was stratified according to number of vaccine doses received at the time of testing antibody: the CS-1st-dose cohort, consisting of individuals who received one dose of the COVID-19 vaccine, and the CS-second-dose cohort, consisting of those who received two doses. Participants were excluded from the CS-1st-dose cohort if they received the first vaccine dose on the day of or more than 84 days (12 weeks) prior to the antibody test. Similarly, participants were excluded from the CS-second-dose cohort if they received the second-dose vaccine on the day of or more than 84 days (12 weeks) before the antibody test. Individuals with evidence of prior infection, as confirmed by the presence of nucleocapsid antibodies, were excluded from both cohorts. To ensure the robustness and reliability of the identified genetic associations, we partitioned participants into discovery (90%) and validation (10%) subsets in a non-random manner. The discovery subset performed genotype calling using the UKBB Axiom array, while the validation subset was processed using the UK BiLEVE Axiom array. This methodological strategy can maximise the independence across both subsets. Notably, our analysis included people who tested positive in the self-test antibody study but did not participate in the coronavirus infection study. Specifically, out of the 81,353 people who tested positive in the self-test antibody study, only 58,821 (72.3%) participated in the follow-up infection study and had a valid anti-N protein result. Of these, 10,783 (18.3%) tested positive for the anti-N protein and were excluded from the study. For the remaining 22,532 (27.6%) individuals who did not participate in the follow-up infection study, and thus had an undetermined prior infection status, we made an assumption that they were COVID-19 naïve and were included in the subsequent analyses. As a result, it is possible that about 5.0% (27.6% × 18.3%) of antibody-positive participants in the current CS cohort could result from the previous infection rather than the vaccination.

The PS cohort included all participants who had a record of COVID-19 vaccination in their linked primary care data between December 1, 2020, and September 30, 2021. The follow-up for the PS cohort began on the date of receipt of the first vaccine dose. Participants with a positive PCR result for COVID-19 before vaccination, as confirmed through data linkage to external data sources, were excluded from the cohort.

**Genotyping and HLA imputation**

Genotyping and initial quality control of the genetic dataset for UKBB participants have previously been documented[24]. Briefly, the genotyping of UKBB participants was undertaken in two phases using two custom-built genome-wide arrays that share 95% of over 820,000 SNP marker content. SNP genotyping was used to impute classical HLA types with four-digit resolution at the MHC class I and the class II regions. The HLA imputation was conducted using a modified HLA*IMP:02 model, which was designed to operate on a multi-population reference panel[33]. Imputation algorithms, tools, and quality control processes can be found online (https://biobank.ctsu.ox.ac.uk/crystal/crystal/docs/HLA_imputation.pdf).

This imputation model performed reasonably well in the entire UKBB sample, with a 4-digit accuracy of HLA alleles ranging from 93.9% for HLA-DRB1 gene to 99.5 for HLA-DPA1 gene among European populations with a posterior probability call threshold of 0.7 (94% of

the UKBB individuals self-reported as White). The utility of the HLA imputation was also confirmed in a previous study by replicating signals of known associations between HLA alleles and 11 self-reported immune-mediated diseases[34].

We grouped people with any of six specific HLA alleles (DQB1*06:01, DQB1*06:02, DQB1*06:03, DQB1*06:04, DQB1*06:05, and DQB1*06:09) as the carrier of HLA-DQB1*06 alleles subtypes. Finally, we examined 203 alleles at the four-digit resolution with frequency ≥1/1000 across 11 HLA genes, including A, B, C, DRB1, DRB3, DRB4, DRB5, DQA1, DQB1, DPA1, and DPB1.

### Ascertainment of antibody-positive and COVID-19 outcomes

The presence of detectable antibodies against SARS-COV-2 among the vaccinated individuals was defined based on a positive antibody result. It was measured by either of the two different LFTs, named the Fortress Fast COVID-19 device and the AbC-19™ rapid test. Fortress Fast COVID-19 home test used among the phase 1 participants is a solid phase immunochromatographic assay for the rapid, qualitative and differential detection of IgG and IgM antibodies to SARS-COV-2 in human whole blood, whereas the AbC-19™ Rapid Test used among the phase 2 participants is single use, in vitro immunochromatographic sandwich assay for the qualitative detection of IgG antibodies against the SARS-CoV-2 trimeric spike protein

Both assays have been validated with a good clinical performance in detecting SARS-COV-2 antibodies: a sensitivity of 98.4% and 98.0% and a specificity of 99.8% and 99.5% for the Fortress Fast COVID-19 device and AbC-19TM rapid test, respectively).

COVID-19 infection status was defined through a combination of data sources, including a positive result on PCR testing, hospital admission with a COVID-19-related diagnosis (the ICD-10 codes of U07.1 and U07.2), or a death certificate in which COVID-19 was listed as the cause of death (using the same ICD-10 code as in hospital admission). Severe cases were defined as those for whom breakthrough infection required hospitalisation or resulted in death.

### Statistical analyses

**Characterisation of vaccinated cohorts.** We evaluated the characteristics of HLA-DQB1*06 carriers and HLA-DQB1*06 non-carriers at baseline, which was when the antibody test was performed for the CS cohort and when PS cohort participants received their 1st-dose of the COVID-19 vaccine. Factors that were potential confounders for the genetic analysis were specified based on prior knowledge, including age, sex, vaccine types (ChAdOx1 and BNT162b2), and time since the latest vaccine dose by week.

**Genome-wide association analysis.** The GWAS of antibody response following COVID-19 vaccination was performed using REGENIE, which is a novel machine-learning-based method and has proved advantageous in controlling for population stratification, relatedness and case-control imbalance[35]. Specifically, we used the Firth logistic regression model and adjusted for baseline age (at the date of antibody testing), sex, genetic batch, and first ten genetic principal components (PCs). We retain only high-quality variants satisfying the following criteria (1) missing call rates ≤1%; (2) minor allele frequency ≥1%; (3) minor allele count ≥ 20; and (4) Hardy−Weinberg equilibrium ≥ $1 \times 10^{-15}$. To further minimise the influence of population stratification, our analysis was restricted to participants of Caucasian ancestry as confirmed by genotype (individuals listed in UKBB data filed 22006).

**HLA gene-based association analysis.** We examined the proportion of antibody positivity among different HLA genotype groups, stratified by time. To quantify the association between HLA alleles and antibody response in the CS cohort, we used a multivariable logistic regression model with each allele of interest as the independent variable (number of copies: 0, 1, 2) and the presence of antibodies (positive or negative)

as the dependent variable. The model for the primary analysis was adjusted for age, sex, self-reported ethnicity, genotyping arrays, and the first ten genetic PCs derived from the entire UKBB population. The significance of HLA alleles in the logistic regression model was tested using the Wald test while accounting for multiple comparisons with false discovery rate (FDR) correction[36]. Associations between HLA alleles and vaccine antibody response were estimated separately in the CS-1-dose and CS-2-dose cohorts. Only alleles with FDR-correct *P* value below 0.05 in both the discovery and validation datasets were considered significant and subsequently analysed. A post-hoc analysis was conducted separately among White and other ethnic populations to assess potential ancestry-dependent effects of HLA alleles.

To investigate the associations between HLA alleles affecting antibody response and the risk of breakthrough infection and severe COVID-19 in real-world settings, we conducted a survival analysis in the PS cohort using the Cox proportional hazard model. Potential confounders were adjusted including age, sex, ethnicity, genotyping arrays, and ten PCs. In our primary analysis, we started follow-up from the first vaccination date until death, the outcome of interest or the end of the study (30th November 2021 before the Omicron outbreak in the UK), whichever occurred first.

**Genetic score.** We constructed a genetic score to measure the combined effect of all HLA allelic variations on the antibody positivity phenotype. The GS was calculated by aggregating the product of the effect size for an allele and the number of allele copies across all HLA genes. Effect size estimates for each allele were coefficients (Beta) derived from the logistic regression model and subsequently modified using three approaches: (GS1) forcing the effect size estimate to zero for all alleles with an FDR-correct *P* value below 0.05, (GS2) forcing the effect size estimate to zero for all alleles with an uncorrected *P* value below 0.05, and (GS3) directly applying the original estimate regardless of the statistical significance. This process of shrinking some coefficients to zero is equivalent to excluding corresponding alleles from the GS calculation.

**Drug/vaccine-target Mendelian randomisation analysis.** We used a Mendelian randomisation analysis to study the causal effect between the exposure of antibody positivity and COVID-19 outcomes. We selected HLA alleles that were statistically significant with antibody response and used them as genetic instrumental variables. The effect size of the instrumental variables with exposures (antibody positivity following the initial vaccine dose and the subsequent dose) and with outcomes (incident breakthrough outcomes) were obtained from our analyses of CS and PS cohorts, respectively. In the primary analysis, we only used independent alleles as instrumental variables. For the sensitivity analysis, all significant alleles were used regardless of their intercorrelation. The analysis was performed using the R packages TwoSampleMR.

### Reporting summary

Further information on research design is available in the Nature Portfolio Reporting Summary linked to this article.

## Data availability

Bonafide researchers can apply for access to individual-level source data used in this study from the UKBB at http://ukbiobank.ac.uk/register-apply/. The aggregated data supporting the findings of this study are available within the paper, its supplementary information, and source data files. Source data are provided with this paper.

## Code availability

All analytic codes used for this study have been deposited in a public GitHub repository (https://github.com/xjq8065524/HLA_COVID_Vax_Antibody_Response).

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

## Acknowledgements

Mr Xie is funded through the Jardine-Oxford Graduate Scholarship and a titular Oxford Clarendon Fund Scholarship. Professor Prieto-Alhambra 's research group has received funding from the European Medicines Agency and Innovative Medicines Initiative. The research was partially supported by the Oxford National Institute for Health and Care Research (NIHR) Biomedical Research Centre. DPA is funded through a NIHR Senior Research Fellowship (Grant number SRF-2018-11-ST2-004). The views expressed in this publication are those of the author(s) and not necessarily those of the NHS, the NIHR or the Department of Health. The authors acknowledge English-language editing by Jennifer A. de Beyer, DPhil (Centre for Statistics in Medicine, University of Oxford), and constructive insights from Dr Hongxiang Zeng and Professor Qizhou Lian on improving the study. The authors express sincere gratitude to all individuals who generously participated in the UKBB study, providing an invaluable resource to advance scientific research. The authors also extend appreciation to the UKBB management team for their dedication and administrative efforts.

## Author contributions

Conceptualisation (Junqing Xie, Chunxiao Li, Xiaoying Zheng, and Daniel Prieto Alhambra); data curation (Junqing Xie); statistical analysis (Junqing Xie, Marta Alcalde Herraiz, Yaqing Gao, Shuo Feng, Shenda Hong,

and Yeda Wu); investigation (Chunxiao Li, Shuo Feng, Jia Wei, Zhuoyao Chen, and Binbin Su); supervision (Annika Jodicke, Xiaoying Zheng, Raghib Ali, Nick Wareham, and Daniel Prieto Alhambra); interpretation of data (Junqing Xie, Beatriz Mothe, Yeda Wu, Catherine Cohet, and Daniel Prieto Alhambra); draughting of the manuscript (Junqing Xie, Chunxiao Li, Yaqing Gao, Shuo Feng, and Jia Wei); and critical revision of the manuscript (Yunhe Wang, Beatriz Mothe, Yu Xu, Catherine Cohet, and Daniel Prieto Alhambra). All authors reviewed and approved the final version. The views expressed in this article are the personal views of the author(s) and may not be understood or quoted as being made on behalf of or reflecting the position of the European Medicines Agency or one of its committees or working parties.

## Competing interests

DPA's department has received grant/s from Amgen, Chiesi-Taylor, Lilly, Janssen, Novartis, and UCB Biopharma. His research group has received consultancy fees from Astra Zeneca and UCB Biopharma. Amgen, Astellas, Janssen, Synapse Management Partners and UCB Biopharma have funded or supported training programmes organised by DPA's department. The remaining authors declare no competing interests.

## Additional information

[1]Centre for Statistics in Medicine and NIHR Biomedical Research Centre Oxford, NDORMS, University of Oxford, Oxford, UK. [2]Infectious Diseases Department, IrsiCaixa AIDS Research Institute, Hospital Universitari Germans Trias i Pujol, Badalona, Spain. [3]Medical Research Council Epidemiology Unit, University of Cambridge, Cambridge, UK. [4]Cambridge Baker Systems Genomics Initiative, Department of Public Health and Primary Care, University of Cambridge, Cambridge, UK. [5]British Heart Foundation Cardiovascular Epidemiology Unit, Department of Public Health and Primary Care, University of Cambridge, Cambridge, UK. [6]Victor Phillip Dahdaleh Heart and Lung Research Institute, University of Cambridge, Cambridge, UK. [7]Nuffield Department of Population Health, Big Data Institute, University of Oxford, Oxford, UK. [8]Oxford Vaccine Group, Department of Paediatrics, University of Oxford, Oxford, UK. [9]Nuffield Department of Medicine, Big Data Institute, University of Oxford, Oxford, UK. [10]Centre for Medicines Discovery, Nuffield Department of Medicine, University of Oxford, Oxford, UK. [11]National Institute of Health Data Science, Peking University, Beijing, China. [12]Institute of Medical Technology, Peking University Health Science Center, Beijing, China. [13]Institute for Molecular Bioscience, The University of Queensland, Brisbane, QLD, Australia. [14]School of Population Medicine and Public Health, Chinese Academy of Medical Sciences/Peking Union Medical College, Beijing, China. [15]Real-World Evidence Workstream, Data Analytics and Methods Task Force, European Medicines Agency, Amsterdam, Noord-Holland, The Netherlands. [16]Public Health Research Center, New York University Abu Dhabi, Abu Dhabi, United Arab Emirates. [17]Department of Medical Informatics, Erasmus University Medical Centre, Rotterdam, The Netherlands. [18]These authors jointly supervised this work: Xiaoying Zheng, Catherine Cohet, Raghib Ali, Nick Wareham. ✉e-mail: daniel.prietoalhambra@ndorms.ox.ac.uk

