## [Peer Review File · Nature Communications]

Relationship between HLA genetic variations, COVID-19 vaccine antibody response, and risk of breakthrough outcomesREVIEWER COMMENTS

Reviewer #1 (Remarks to the Author):

General:

In this study, the authors conducted a population-based genomic study to replicate prior findings where HLA-DQB1:06 was associated with increased SARS-CoV-2-specific antibody response following vaccination and reduced breakthrough infection. Second, they identified novel HLA alleles associated with antibody responses. Finally, they estimated vaccine effectiveness against infection and severe disease attributable to vaccine-induced antibodies. The authors should be congratulated for their efforts in trying to clarify whether HLA variability plays a significant role in influencing SARS-CoV-2-specific antibody generation at a population level. This work makes a timely and highly relevant contribution to the ongoing debate on the role of host genomic determinants of immune responses against SARS-CoV-2. To improve the quality of the manuscript, I have forwarded specific major and minor comments below:

Major:

1. In the methods section, under ascertainment of antibody positive and COVID-19 outcomes, the antibody assay employed for the association with the HLA variants is vaguely described. This is a major shortcoming of the study design. Is the assay used a lateral flow? How can one ascertain that the antibody generated was specific to the administered SARS-CoV-2 vaccines? Paradoxically, in the results text, it is indicated that 28.4% of recipients tested positive for anti-spike SARS-CoV-2 antibodies which escalated to 65.5% among individuals who had the antibody test a median of 20 days after administration of the second vaccine dose. Is this anti-Spike antibody assay a lateral flow? The reviewer was able to ascertain that the AbC-19TM Rapid Test can detect the IgG antibodies against the SARS-CoV-2 trimeric spike protein. However, there is no info on whether Fortress Fast COVID-19 Device can do the same. It should be clear to the reader whether all included individuals were tested by the AbC-19TM Rapid Test. The authors should provide explicit details on the type of antibody tests used in their work in the methods section.

My main concern is the fact that the investigators solely base their conclusions on the association between HLA variability and qualitative antibody response, and there is a lack of data on quantitative (for example geometric mean titer) antibody response. The major difference between the current work and the one by Mentzer et al (Nat Med 2023;29:147) was the fact that the latter study determined antibody titer – hence the differences between the two studies. In addition, a positive antibody result may not suffice, and the titer is more important. All of these were not reflected in the discussion section.

In the same token, in the methods section under nested COVID-19 infection seroprevalence study in UKBB, the authors note that antibody-positive participants were re-invited to provide a capillary blood sample for laboratory analysis of specific antibodies (i.e., nucleocapsid) that are solely produced in response to COVID-19 infection. Please provide the name/brand of the kit used to detect the anti-nucleocapsid antibody.

2. In the discussion section, the authors note that the HLA-A03:01 allele was also associated with an enhanced antibody response and noted that this observation may reflect the epidemiological evidence of a correlation between reactogenicity and immunogenicity following COVID-19 vaccination. This phenomenon has already been described recently by Italian investigators (Magri et al. HLA 2023 Jul 19, doi:10.1111/tan.15157) demonstrating that increased antibody levels and side effects in carriers of the HLA-A*03:01 allele.

Given that HLA class I molecules are basically responsible for CTL responses, can the authors elaborate on this further? Is the association between HLA-A03:01 and antibody production reported in this study and others coincidental? Perhaps the link between HLA-A03:01 and reactogenicity/immunogenicity is mediated through cytotoxic mechanisms rather than increased antibody production (Ref: Viruses. 2023;15:906. doi:10.3390/v15040906). Please provide comments on this in the discussion section.

3. For the genetic analysis, the authors included factors that were potential confounders specified a priori. These were age, sex, vaccine type, and time-lapse since the latest vaccine dose. Nonetheless, other significant confounders that impact antibody production were left out. For example, co-morbid conditions (Nat Microbiol 2021; 1–10. doi:10.1038/s41564-021-00947-3). Please comment on this.

Minor:

1. Within the results section, please correct Figure a to Figure f throughout as Figure 1a to Figure 1f.

2. Figure 2 title indicates 203 HLA alleles while the text in the results section (under Discovery and validation of novel HLA allele associations with antibody response) shows 213 HLA alleles. Check if this is a typo! Throughout the manuscript as well!

3. There are two Figure 2s. Please correct and change the 2nd Figure 2 as Figure 3, and all the subsequent figures need to be corrected.

4. Bertinello et al (HLA 2023;102:301) reported an association between A*03:01, B*40:02 and DPB1*06:01 and high antibody concentration, and between A*24:02, B*08:01 and C*07:01 and low humoral responses. Please discuss their findings with that of the available evidence, including your findings.

5. There are only two published GWAS studies assessing the association of antibody response to HLA variability (Mentzer et al (Nat Med 2023;29:147; Magri et al. HLA 2023 Jul 19, doi:10.1111/tan.15157). In the discussion section, under "Findings in context", the authors referred few studies – all are not based on GWAS. Please modify the statement in the text.

6. In the discussion section, under Findings in the Context, the authors noted that there is a clinical agreement that vaccination is efficacious and offers superior protection against severe COVID-19 than infection. This consensus is based on evidence gathered from initial vaccine trials as well as routinely collected health data. Please provide references for this!

7. The findings support the notion that HLA genes modulate the response to COVID-19 vaccines and highlight the need for genetic studies in diverse populations. Your analysis was restricted to white ancestry only (to reduce population stratification). Please comment on this in the discussion text!

Reviewer #2 (Remarks to the Author):

In this paper the authors present an important and comprehensive study to investigate the genetic determinants of antibody response to COVID-19 vaccines and their impact in infection and hospitalisation.

However I think the manuscript will benefit from a lot more clarity on what exactly they are doing and how.

1-The authors claim that they have done a GWAS but present only results from the HLA data. Was there a genome-wide analysis done and the only significant signals are in HLA or was only HLA analysed (which is not a GWAS)? It is not clear what was the model used for the analysis, was it a linear mixed model or a logistic model?

2-The authors use individuals with only Caucasian ancestry for the genetic analysis but do not explain how was it defined. Was it self-reported ancestry? in which case it should be validated by genotype, or are they using individuals with European or white British ancestry confirmed by genotype? In any case, this is a big limitation of the study, and should be noted because findings would have to be replicated in other ancestries (I understand UK Biobank resource may not have the numbers to do a comprehensive genetic study in non-European ancestries).

3- In the genetic study authors divide the CS cohort into discovery and validation. Do they validate their genetic findings? How do they do that, do they set a p-value threshold or do they only use it for the gene level analysis afterwards?

My understanding is that the authors use the discovery and validation cohorts to create the genetic score, how do they validate it then? The study would largely benefit from an external test at this level.

4- In the gene level analysis, the authors say that they correct for ethnicity. Again, is this genetically confirmed ancestry? Or do they correct for self-reported ethnicity which has a big genetic error? Do they add non-caucasian individuals in the analysis then? It is not clear from the methods or figure 1 at all.

5- The authors claim that there are 7 clustered signals with highly correlated snps, but they also claim causal alleles from these clusters. Is it just the allele with lower p-val from each cluster the one assumed to be causal?

6- The authors have a Cross-sectional cohort and a Prospective cohort but they seem to overlap. And then they do a two sample MR, how do they insure independence of the datasets? It seems that they use the same cohort (UKB) for exposure and output which is just one sample. I think one sample MR would be more appropriate unless they do separate the cohorts in two.

Minor comments

- In figure 1 there seems to be more than one independent signal, while when talking about results at the beginning only the snp with lowest p-value is mentioned. If there are independent signals, how many in which analysis, the lead snp and the p-val should be mentioned.

- Figure a) figure e) means figure 2 a) figure 2 e)

- Quality of Figures, figure 2 labels difficult to read.

- There are 2 figures 2

- Manhattan and circular manhattan do not give different information?

Reviewer #3 (Remarks to the Author):

Xie et al., performed a GWAS with antibody positivity on a very large cohort of individuals from the UK. A similar study has been performed, by Mentzer et al., 2023, albeit on a smaller sample size. The strength of the current study is the size of the cohort, however I feel there are also some important limitations.

Antibody positivity was determined using an at home lateral flow test. Whilst the reported sensitivity and specificity of the LFT is reported at around 98%, the self-reported antibody positivity rates following first and second vaccination in this cohort seem very low (28.4% and 65.5%) compared to previously published reports, even for an aged cohort. Therefore a substantial number of individuals with true antibody positivity may have been missed which would likely confound the results with regards to the HLA types identified as well as the estimates of antibody effectiveness and is a major drawback of this work. A strength of the previously published Mentzer et al paper was that it was able to perform an association between antibody titres and HLA types rather than just antibody positivity.

The methods also state that individuals were excluded from the study if they received their vaccine on the day of the antibody test, implying that individuals who received a vaccination and then performed the test within a few days were still included, and given that antibody responses are known to take at least several days to develop, this may have contributed to the higher than expected antibody negative individuals.

The discussion refers to work performed by Cromer et al., 2023 stating this work showed that "antibodies may provide approximately 20% protection against infection and 50% protection against severe COVID-19." I don't believe that claim is made by the Cromer paper which reports

that the lowest reported/detectable titres of neutralising antibodies are “20% of convalescent” which would give efficacies of around 50% and 80% for symptomatic and severe disease respectively, based on the models in that paper. Furthermore, the data from the Comer paper data is based on neutralising antibodies that only make up a fraction of the total antibody pool, so results using total antibodies, as done in this current study, would be expected to be higher.

The discussion also states that this work will help address vaccine hesitancy, I am not sure how that would happen?

There are also many typographical errors in the manuscript that need addressing, including, but not limited to:

There are 2 figures labelled figure 2, one should be figure 3. The real figure 2 is not referred to correctly in the manuscript, ie it just says ‘figure a’.

What do the error bars on figure 2B represent?

In the second paragraph of the discovery and validation paragraph, the second sentence should read ‘Among them 13 HLA alleles....response, 1 with second-dose’ rather than 14 and 2, otherwise the numbers don’t add up to 15.

The last paragraph of the introduction should read “SARS-CoV-2 breakthrough infection” not COVID-19.

Line 5 of the results should read “as confirmed by the absence of antibodies against SARS-CoV-2 nucleocapsid antibody”?

REVIEWER COMMENTS

Reviewer #1 (Remarks to the Author):

General:

In this study, the authors conducted a population-based genomic study to replicate prior findings where HLA-DQB1:06 was associated with increased SARS-CoV-2-specific antibody response following vaccination and reduced breakthrough infection. Second, they identified novel HLA alleles associated with antibody responses. Finally, they estimated vaccine effectiveness against infection and severe disease attributable to vaccine-induced antibodies. The authors should be congratulated for their efforts in trying to clarify whether HLA variability plays a significant role in influencing SARS-CoV-2-specific antibody generation at a population level. This work makes a timely and highly relevant contribution to the ongoing debate on the role of host genomic determinants of immune responses against SARS-CoV-2. To improve the quality of the manuscript, I have forwarded specific major and minor comments below:

Major:

1. In the methods section, under ascertainment of antibody positive and COVID-19 outcomes, the antibody assay employed for the association with the HLA variants is vaguely described. This is a major shortcoming of the study design. Is the assay used a lateral flow? How can one ascertain that the antibody generated was specific to the administered SARS-CoV-2 vaccines? Paradoxically, in the results text, it is indicated that 28.4% of recipients tested positive for anti-spike SARS-CoV-2 antibodies which escalated to 65.5% among individuals who had the antibody test a median of 20 days after administration of the second vaccine dose. Is this anti-Spike antibody assay a lateral flow? The reviewer was able to ascertain that the AbC-19™ Rapid Test can detect the IgG antibodies against the SARS-CoV-2 trimeric spike protein. However, there is no info on whether Fortress Fast COVID-19 Device can do the same. It should be clear to the reader whether all included individuals were tested by the AbC-19™ Rapid Test. The authors should provide explicit details on the type of antibody tests used in their work in the methods section.

Response:

We want to thank the reviewer for this valuable feedback. Based on their suggestion, We have now expanded the methodological details in the manuscript as requested, and offer a concise summary here for ease of peer-review (please refer to track changes in **pages 23-25** of the updated manuscript for the full detail).

Our study used the Fortress Fast COVID-19 Device alongside the AbC-19™ Rapid Test. Specifically for the former, it was designed for both professional and lay use of self-testing. It is a lateral flow assay that qualitatively detects SARS-CoV-2 IgG and IgM antibodies in capillary blood, serum, or plasma. Further product details are available at official website. <https://unahealth.co.uk/product/fortress-diagnostics-covid-19-rapid-antibody-testing/#product-details>.

My main concern is the fact that the investigators solely base their conclusions on

the association between HLA variability and qualitative antibody response, and there is a lack of data on quantitative (for example geometric mean titer) antibody response. The major difference between the current work and the one by Mentzer et al (Nat Med 2023;29:147) was the fact that the latter study determined antibody titer – hence the differences between the two studies. In addition, a positive antibody result may not suffice, and the titer is more important. All of these were not reflected in the discussion section.

Response:

Thank you for highlighting the distinction between qualitative and quantitative antibody responses.

We would firstly like to stress that based on our opinion (and that of the consulted clinicians including our co-authors), the qualitative antibody response used in our study signals minimum antibody levels post-vaccination, and is therefore a phenotype of clinical significance. While we agree with the reviewer that titres provide additional information, our approach remains relevant, and is more likely in line with testing practices in actual clinical settings during the pandemic. For a large-scale population-based study like ours, including hundreds of thousands of participants, assessing qualitative antibody responses is the most feasible method. This is the reason why the UK Biobank decided to conduct this test.

Secondly, although binary antibody response analyses may be statistically less efficient than continuous measures like titres, our study was over 200x times larger in sample size compared to the previous (e.g. ~200,000 participants in ours vs ~1,000 in Mentzer et al.'s study), therefore compensating for this limitation.

As demonstrated by our findings, the considerable scale of our study allowed us to identify a wider array of HLA alleles affecting COVID-19 antibody response, which includes and extends beyond the scope of Mentzer et al.'s research.

Despite the above, we agree with the reviewer that more text should be added on this, and we have discussed this point as a potential study limitation in **page 8**.

“The genetic profile for the binary phenotype of anti-S antibody seropositivity is not as statistically efficient as continuous measures like antibody titres and may not fully encapsulate other immune attributes such as peak antibody titre, durability, and specificity. Yet, our study size was over two hundred-fold larger than the second largest study on this topic, therefore compensating for this limitation and allowing for the identification of more relevant HLA alleles affecting COVID-19 antibody response, including and extending the previously reported ones”.

In the same token, in the methods section under nested COVID-19 infection seroprevalence study in UKBB, the authors note that antibody-positive participants were re-invited to provide a capillary blood sample for laboratory analysis of specific antibodies (i.e., nucleocapsid) that are solely produced in response to COVID-19 infection. Please provide the name/brand of the kit used to detect the anti-nucleocapsid antibody.

Response:

The Elecsys® Anti-SARS-CoV-2 immunoassay was used to detect the presence of high affinity antibodies against SARS-CoV-2 nucleocapsid proteins. More details about this product can be found at the official website.

<https://diagnostics.roche.com/gb/en/products/params/electsys-anti-sars-cov-2.html>.

Please see track changes in **page 24**.

2. In the discussion section, the authors note that the HLA-A03:01 allele was also associated with an enhanced antibody response and noted that this observation may reflect the epidemiological evidence of a correlation between reactogenicity and immunogenicity following COVID-19 vaccination. This phenomenon has already been described recently by Italian investigators (Magri et al. HLA 2023 Jul 19, doi:10.1111/tan.15157) demonstrating that increased antibody levels and side effects in carriers of the HLA-A*03:01 allele.

Response:

Thanks for providing us this relevant article, which we have incorporated it in our discussion. Please see track changes in **page 7-8**.

Given that HLA class I molecules are basically responsible for CTL responses, can the authors elaborate on this further? Is the association between HLA-A03:01 and antibody production reported in this study and others coincidental? Perhaps the link between HLA-A03:01 and reactogenicity/immunogenicity is mediated through cytotoxic mechanisms rather than increased antibody production (Ref: Viruses. 2023;15:906. doi:10.3390/v15040906). Please provide comments on this in the discussion section.

Response:

The mechanisms behind the association between the HLA-A*03:01 and higher seropositivity is unclear and warrants further specific functional studies, but it is intriguing because its recent additional association with higher reactogenicity. One potential biological pathway stands on the cross-reactive highly restricted HLA-A*03:01 hCoV-specific CD8 T cells, which exists prior to SARS-CoV-2 exposure to facilitate a higher response upon vaccination as well as side effects. Please see track changes in **page 7-8**.

3. For the genetic analysis, the authors included factors that were potential confounders specified a priori. These were age, sex, vaccine type, and time-lapse since the latest vaccine dose. Nonetheless, other significant confounders that impact antibody production were left out. For example, co-morbid conditions (Nat Microbiol 2021; 1–10. doi:10.1038/s41564-021-00947-3). Please comment on this.

Response:

We sincerely value the reviewer's comment regarding the impact of chronic diseases on vaccine antibody responses. Nonetheless, defining their role as confounders in our research context may be challenging, given the absence of a demonstrated correlation of them with these HLA genetic variations (not fulfil the confounder definition).

Additionally, certain chronic conditions, such as rheumatoid arthritis and asthma as reported in the reviewer's mentioned paper, could potentially be components of the causal pathway connecting HLA mutations to antibody responses.

These considerations have deterred us to adjust for comorbidities in our analysis.

Minor:

1. Within the results section, please correct Figure a to Figure f throughout as Figure 1a to Figure 1f.

Response:

Thanks. We have corrected this. Please see track changes in **page 4-5**.

2. Figure 2 title indicates 203 HLA alleles while the text in the results section (under Discovery and validation of novel HLA allele associations with antibody response) shows 213 HLA alleles. Check if this is a typo! Throughout the manuscript as well!

Response:

Thanks. It was a typo and we have corrected it.

3. There are two Figure 2s. Please correct and change the 2nd Figure 2 as Figure 3, and all the subsequent figures need to be corrected.

Response:

Thanks. We have corrected this.

4. Bertinello et al (HLA 2023;102:301) reported an association between A*03:01, B*40:02 and DPB1*06:01 and high antibody concentration, and between A*24:02, B*08:01 and C*07:01 and low humoral responses. Please discuss their findings with that of the available evidence, including your findings.

Response:

Thanks, we have added this most recent result in Discussion. Please see track changes in **page 7-8**.

5. There are only two published GWAS studies assessing the association of antibody response to HLA variability (Mentzer et al (Nat Med 2023;29:147; Magri et al. HLA 2023 Jul 19, doi:10.1111/tan.15157). In the discussion section, under "Findings in context", the authors referred few studies – all are not based on GWAS. Please modify the statement in the text.

Response:

Thanks. We have modified the discussion as suggested. Please see track changes in **page 7**.

6. In the discussion section, under Findings in the Context, the authors noted that there is a clinical agreement that vaccination is efficacious and offers superior protection against severe COVID-19 than infection. This consensus is based on

evidence gathered from initial vaccine trials as well as routinely collected health data. Please provide references for this!

Response:

Thanks. We have added the references.

7. The findings support the notion that HLA genes modulate the response to COVID-19 vaccines and highlight the need for genetic studies in diverse populations. Your analysis was restricted to white ancestry only (to reduce population stratification). Please comment on this in the discussion text!

Response:

Thanks. We have acknowledged this could be a potential limitation of our study. Please see track changes in **page 9**.

Reviewer #2 (Remarks to the Author):

In this paper the authors present an important and comprehensive study to investigate the genetic determinants of antibody response to COVID-19 vaccines and their impact in infection and hospitalisation.

However I think the manuscript will benefit from a lot more clarity on what exactly they are doing and how.

Response:

We thank the reviewer for the positive comments for our study and suggestions to improve the reporting of our results.

1-The authors claim that they have done a GWAS but present only results from the HLA data. Was there a genome-wide analysis done and the only significant signals are in HLA or was only HLA analysed (which is not a GWAS)? It is not clear what was the model used for the analysis, was it a linear mixed model or a logistic model?

Response:

We performed a genome-wide analysis across all genomes. However, we specifically focused on presenting the results for Chromosome 6, the location of the HLA genes, to align with our study's primary objective. To avoid potential confusion, we have reported both the genome-wide and region-specific statistical significances, which are based on varying counts of association tests (Please refer to the threshold lines in **Figure 2A** and **2E**). The entire GWAS results have also been added as a **supplement 1** in the updated manuscript.

We used Firth logistic regression modelling for the GWAS.

2-The authors use individuals with only Caucasian ancestry for the genetic analysis but do not explain how it was defined. Was it self-reported ancestry? in which case it should be validated by genotype, or are they using individuals with European or white British ancestry confirmed by genotype? In any case, this is a big limitation of the study, and should be noted because findings would have to be replicated in other ancestries (I understand UK Biobank resource may not have the numbers to do a comprehensive genetic study in non-European ancestries).

Response:

In our study, the identification of Caucasian ethnicity was not based on self-reporting but determined through genetic principal components analysis conducted by the UK Biobank team, which has been commonly used in prior research (e.g. <https://www.ncbi.nlm.nih.gov/pmc/articles/PMC6969355/>).

Also, it is important to note that the GWAS analysis, while included, was not the central focus of our study. It was conducted only to validate a prior theoretical hypothesis: that SNP-level variations in MHC regions should be correlated with antibody responses in our dataset. Therefore, for this specific analysis, we prioritized internal validity of the findings and restricted our cohort to a more homogeneous population.

However, our primary focus of HLA gene-level associations with antibody responses includes more diverse ethnic groups, as reported in **Extended Table 1** of our manuscript.

Nonetheless, we acknowledge that the majority of UK Biobank participants are of White ancestry, a limitation we have noted in our discussion section (Please see track changes in **page 9**).

3- In the genetic study authors divide the CS cohort into discovery and validation. Do they validate their genetic findings? How do they do that, do they set a p-value threshold or do they only use it for the gene level analysis afterwards? My understanding is that the authors use the discovery and validation cohorts to create the genetic score, how do they validate it then? The study would largely benefit from an external test at this level.

Response:

We have indeed validated the discovered novel HLA allele and applied the False Discovery Rate (FDR) method to account for multiple testing. Please refer to **Extended Table 3** and the Methods (HLA gene-based association analysis **page 27**) in our manuscript.

Given the absence of an external dataset for validating the genetic score for the antibody response outcome, we conducted an extrapolation study to prove the score's validity and utility for the breakthrough infection outcome, which we consider to be of greater clinical significance.

Please see extrapolation results in **Page 6**.

4- In the gene level analysis, the authors say that they correct for ethnicity. Again, is this genetically confirmed ancestry? Or do they correct for self-reported ethnicity which has a big genetic error? Do they add non-caucasian individuals in the analysis then? It is not clear from the methods or figure 1 at all.

Response:

We adjusted self-reported ethnicity and principal genetic components (PC) in the gene level analysis. We are concerned that adjusting genetically confirmed ancestry will result in overadjustment issue as PC and genetic ancestry are highly corrected.

Please see track changes in **page 27**.

5- The authors claim that there are 7 clustered signals with highly correlated snps, but they also claim causal alleles from these clusters. Is it just the allele with lower p-val from each cluster the one assumed to be causal?

Response:

Yes, we have clarified this point in the abstract. Please see track changes in **page 2**.

6- The authors have a Cross-sectional cohort and a Prospective cohort but they

seem to overlap. And then they do a two sample MR , how do they insure independence of the datasets? It seems that they use the same cohort (UKB) for exposure and output which is just one sample. I think one sample MR would be more appropriate unless they do separate the cohorts in two.

Response:

Indeed, there was an overlap in the samples used for the HLA-antibody response and HLA-COVID infection associations. Despite this, we opted for the two-sample Mendelian Randomization (MR) approach due to its greater flexibility in testing a range of underlying assumptions compared to the one-sample MR method.

We acknowledge the reviewer's point about the literature recommendation for the independence of datasets in two-sample MR analysis. However, recent empirical studies suggest that this may not be essential, and show that two-sample methods can yield valid results even with overlapping samples, particularly in large cohorts such as ours (see below reference paper for details).

- The use of two-sample methods for Mendelian randomization analyses on single large datasets (<https://pubmed.ncbi.nlm.nih.gov/33899104/>)

Minor comments

- In figure 1 there seems to be more than one independent signal, while when talking about results at the beginning only the snp with lowest p-value is mentioned. If there are independent signals, how many in which analysis, the lead snp and the p-val should be mentioned.

Response:

Thank you. As previously mentioned, the primary focus of our study is the identification of independent HLA genes rather than independent SNPs.

- Figure a) figure e) means figure 2 a) figure 2 e)
- Quality of Figures, figure 2 labels difficult to read.
- There are 2 figures 2
- Manhattan and circular manhattan do not give different information?

Response:

Thanks. We have corrected all of them.

As for the last point, the reason why we chose to use circular Manhattan is to save the plot space.

Reviewer #3 (Remarks to the Author):

Xie et al., performed a GWAS with antibody positivity on a very large cohort of individuals from the UK. A similar study has been performed, by Mentzer et al., 2023, albeit on a smaller sample size. The strength of the current study is the size of the cohort, however I feel there are also some important limitations.

Antibody positivity was determined using an at home lateral flow test. Whilst the reported sensitivity and specificity of the LFT is reported at around 98%, the self-reported antibody positivity rates following first and second vaccination in this cohort seem very low (28.4% and 65.5%) compared to previously published reports, even for an aged cohort. Therefore a substantial number of individuals with true antibody positivity may have been missed which would likely confound the results with regards to the HLA types identified as well as the estimates of antibody effectiveness and is a major drawback of this work. A strength of the previously published Mentzer et al paper was that it was able to perform an association between antibody titres and HLA types rather than just antibody positivity.

The methods also state that individuals were excluded from the study if they received their vaccine on the day of the antibody test, implying that individuals who received a vaccination and then performed the test within a few days were still included, and given that antibody responses are known to take at least several days to develop, this may have contributed to the higher than expected antibody negative individuals.

Response:

The lower antibody positivity rates in our cohorts (28.4% and 65.5%) compared to previous literature are less likely due to misclassification but more likely are attributable to including a significant number of individuals recently vaccinated prior to antibody testing. This is somewhat related to the reviewer's second comment.

Rather than merely presenting overall antibody positivity, we have detailed the results by weeks (refer to **Figure 2B/2F**). At the 4th week, antibody positivity was approximately 40% and 80% following the first and second vaccinations, aligning closely with previous studies, especially considering the older age of our cohorts as noted in the reviewer's first comment.

Furthermore, we agree in general with the reviewer that "antibody responses are known to take at least several days to develop". However, applying this uniformly to a large population lacks scientific rigor. Antibody development is a continuous process and have varying speed between individuals, and it is likely a proportion of individuals can show positive antibody results even in earlier days after vaccination, as shown in our **Figure 2B/2F** within 1 week.

The discussion refers to work performed by Cromer et al., 2023 stating this work showed that "antibodies may provide approximately 20% protection against infection and 50% protection against severe COVID-19." I don't believe that claim is made by the Cromer paper which reports that the lowest reported/detectable titres of neutralising antibodies are "20% of convalescent" which would give efficacies of

around 50% and 80% for symptomatic and severe disease respectively, based on the models in that paper. Furthermore, the data from the Comer paper data is based on neutralising antibodies that only make up a fraction of the total antibody pool, so results using total antibodies, as done in this current study, would be expected to be higher.

Response:

The statement in our discussion section is based on **Figure 1A** in the referenced paper, where the left end of the axis indicates vaccine efficacies of 20% for infection and 50% for severe outcomes.

We have revised the text in our manuscript to clarify the points highlighted by the reviewer. Please see track changes in **page 8**.

Cromer, D., Steain, M., Reynaldi, A. et al. Predicting vaccine effectiveness against severe COVID-19 over time and against variants: a meta-analysis. *Nat Commun* 14, 1633 (2023)

The discussion also states that this work will help address vaccine hesitancy, I not sure how that would happen?

Response:

We have revised the statement to “Our research is the first application of Mendelian randomization to estimate COVID-19 vaccine effectiveness. This additional layer of genetic evidence supporting the protective role of vaccine-induced antibodies can further improve confidence in COVID-19 vaccines and tackle vaccine hesitancy.” Please see track changes in **page 8**.

There are also many typographical errors in the manuscript that need addressing, including, but not limited to:

There are 2 figures labelled figure 2, one should be figure 3. The real figure 2 is not referred to correctly in the manuscript, ie it just say 'figure a'.

What do the error bars on figure 2B represent?

In the second paragraph of the discovery and validation paragraph, the should the second sentence read 'Among them 13 HLA alleles....response, 1 with second-dose" rather than 14 and 2, otherwise the numbers don't add up to 15.

The last paragraph of the introduction should read "SARS-CoV-2 breakthrough infection" not COVID-19.

Line 5 of the results should read "as confirmed by the absence of antibodies against SARS-CoV-2 nucleocapsid antibody"?

Response:

We thank the reviewer for their valuable feedback. All the points raised have been corrected and incorporated into the revised manuscript.

REVIEWER COMMENTS

Reviewer #1 (Remarks to the Author):

The investigators provided more info regarding the Fortress FAST COVID-19 antibody test detecting IgG/IgM antibodies. It is not justified why it was included. Other than neutralizing antibody, anti Spike antibody is a good correlate of COVID-19 vaccine response. The authors should limit analysis of antibody response solely based on the Abc-19 rapid test results only as this LFA kit detects anti Spike antibody.

Gilbert et al (Science 2021; 375:43) demonstrated explicitly that the magnitude of vaccine efficacy correlated with levels of binding and neutralizing antibodies against the viral spike protein. The higher the antibody level, the greater the protection afforded by the mRNA vaccine. A UK study noted that vaccine efficacy of 80% against primary symptomatic COVID-19 was achieved with anti-Spike IgG level of 40,923 arbitrary units (AU)/mL (Nat Med 2021;27:2032). Hence, the argument that someone can compensate titers by increasing sample size is not justified. The investigators should discuss these details, in particular, lack of anti Spike titer as well as lack of neutralization titer data, which is the major limitation of the current (kind of retrospective) data analysis.

Finally, in this and the study done by the Italians, the HLA-A*03:01 is associated with enhanced antibody response. My question was - is this a coincidental finding? We all know, from basic immunology principles, that it is the HLA class II that is relevant in the generation of antibodies through their effects on T follicular helper cells. A recent review (Viruses. 2023;15:906) showed that almost all studies revealed no association between HLA class I and antibody response. The authors did not discuss on this and they should revisit it.

Reviewer #1 (Remarks on code availability):

Sufficient.

Reviewer #2 (Remarks to the Author):

I thank the authors for their replies to my comments, particularly for pointing me towards the evidence that two sample MR may be sufficient in large studies. I find the manuscript to be improved but I still have a concern.

I think the authors should be very mindful of their language, ethnicity is cultural and, although it is correlated, it is not a genetic term (or something that can be inferred from genetics). When talking about "Caucasian ethnicity confirmed by genotype" they probably could refer to "European ancestry defined by genotype" (or white British depending on which one they are using). LD in the HLA region is more complex than in the autosome, and therefore, ancestry is more important in HLA than in the autosomal regions. When doing the analysis, I am not convinced that correcting for ethnicity and the 10 PCs is enough. Ideally, the analysis should be done separately by ancestry and then meta-analyse to create a trans-ancestry model, but at least the authors should show that the effects are not ancestry-dependant showing ancestry-specific effects.

Reviewer #2 (Remarks on code availability):

The code is not there, there is just a code for vaccine effectiveness. Probably will be there on publication.

Reviewer #3 (Remarks to the Author):

The authors have addressed my previous concerns except for the interpretation of Fig 1A from Cromer et al., 2023. In figure 1A provided in the rebuttal by the authors, the point at which the curves touch the y-axis is well below the level of detection of neutralising antibodies in most assays- hence why there are no data points this low on the curve, the modelling extrapolated the curves to this point. As in my previous comment, the lowest reported/detectable levels of neutralising antibodies are ~20% of convalescent titres, or somewhere between 0.125 and 0.25 on the x-axis of Fig 1a, corresponding to protection efficacies of 50% and 80% for symptomatic and severe disease respectively.

The discussion should be amended to reflect this.

REVIEWER COMMENTS

Reviewer #1 (Remarks to the Author):

The investigators provided more info regarding the Fortress FAST COVID-19 antibody test detecting IgG/IgM antibodies. It is not justified why it was included. Other than neutralizing antibody, anti Spike antibody is a good correlate of COVID-19 vaccine response. The authors should limit analysis of antibody response solely based on the AbC-19 rapid test results only as this LFA kit detects anti Spike antibody.

Response:

We sincerely thank the reviewer for insightful comments and apologize for the ambiguity may present in our manuscript. To clarify:

In our study, the recruitment of participants for antibody testing was conducted in two distinct phases. During the first phase, participants were tested using the Fortress Fast COVID-19 Test, whereas in the second phase, the AbC-19™ Rapid Test was used. We have elaborated on these procedures in the Methods section (please refer to page 24, lines 32-38, and page 26, lines 107-110).

Although we acknowledge the reviewer's interest in targeting the anti-Spike antibody specifically, our study's primary aim was broader, aiming to detect any antibodies against SARS-CoV-2, not limited to a specific type.

Yet, considering that the only two vaccines (Pfizer and AstraZeneca) administered in the UK Biobank participants are both designed to elicit a human immune response primarily targeting the spike (S) rather than nucleocapsid (N) protein, we infer that the measured outcome is likely more pertinent to the anti-Spike antibody, even though the Fortress Fast COVID-19 Test, like most rapid antibody tests, does not specifically differentiate between the S and N proteins, the two major antigens of SARS-CoV-2.

Gilbert et al (Science 2021; 375:43) demonstrated explicitly that the magnitude of vaccine efficacy correlated with levels of binding and neutralizing antibodies against the viral spike protein. The higher the antibody level, the greater the protection afforded by the mRNA vaccine. A UK study noted that vaccine efficacy of 80% against primary symptomatic COVID-19 was achieved with anti-Spike IgG level of 40,923 arbitrary units (AU)/mL (Nat Med 2021;27:2032). Hence, the argument that someone can compensate titers by increasing sample size is not justified. The investigators should discuss these details, in particular, lack of anti Spike titer as well as lack of neutralization titer data, which is the major limitation of the current (kind of retrospective) data analysis.

Response:

We thank the suggestion and have added this limitation in the Discussion section as below (Please refer to page 8-9):

Also, the presence of antibody provides little information on other immune attributes, such as antibody's peak titre, durability, specificity and neutralizing capability, and cannot directly infer the protective efficacy of the detected antibodies.

Finally, in this and the study done by the Italians, the HLA-A*03:01 is associated with enhanced antibody response. My question was - is this a coincidental finding? We all know, from basic immunology principles, that it is the HLA class II that is relevant in the generation of antibodies through their effects on T follicular helper cells. A recent review (Viruses. 2023;15:906) showed that almost all studies revealed no association between HLA class I and antibody response. The authors did not discuss on this and they should revisit it.

Response:

We thank the comment. However, we may disagree with the notion that the basic principles of immunology, which highlight the importance of HLA class II in antibody generation, would entirely preclude the possibility that variations in HLA class I genes can also influence antibody responses either through direct or indirect pathways.

Firstly, our study, **along with an independent Italian study**, found that the class I HLA-A*03:01 allele was associated with the COVID-19 antibody response. Furthermore, this allele has been linked to COVID-19 vaccine-related side effects **in another large genome-wide association study (GWAS)**. The convergence of evidence from these studies strongly suggests that the HLA-A*03:01 finding is not merely coincidental. For a detailed discussion, please refer to our discussion section (line 257-271).

Regarding few associations between HLA alleles and antibody response has been found in the reviewer mentioned literature, we suppose it is likely due to those prior published studies only involved a small sample of participants, with the majority less than 100 individuals, which leads to insufficient statistical power to detect an effect, rather than an absence of biological relevance of HLA variations.

Reviewer #1 (Remarks on code availability):

Sufficient.

Response:

Thanks for the positive feedback.

Reviewer #2 (Remarks to the Author):

I thank the authors for their replies to my comments, particularly for pointing me towards the evidence that two sample MR may be sufficient in large studies. I find the manuscript to be improved but I still have a concern.

Response:

Thanks for the positive feedback.

I think the authors should be very mindful of their language, ethnicity is cultural and, although it is correlated, it is not a genetic term (or something that can be inferred from genetics). When talking about "Caucasian ethnicity confirmed by genotype" they probably could refer to "European ancestry defined by genotype" (or white British depending on which one they are using).

LD in the HLA region is more complex than in the autosome, and therefore, ancestry is more important in HLA than in the autosomal regions. When doing the analysis, I am not convinced that correcting for ethnicity and the 10 PCs is enough. Ideally, the analysis should be done separately by ancestry and then meta-analyse to create a trans-ancestry model, but at least the authors should show that the effects are not ancestry-dependant showing ancestry-specific effects.

Response:

We thank the reviewer for the comment.

Firstly, defining Caucasian ethnicity by genotype within the UK Biobank has been a common and widely accepted practice in prior research (doi: 10.1093/hmg/ddz175).

Also, the decision on whether correcting for ethnicity and the 10 PCs is sufficient or not should be viewed in the specific context. For example, we acknowledge that the HLA is among the most polymorphic regions in the human genome. However, this characteristic does not necessarily imply a confounding bias in epidemiological studies involving HLA, particularly since we found no evidence that ethnicity is correlated with the outcome we measured: antibody response.

Lastly, using a trans-ancestry model as proposed by the reviewer is promising, but, in this study, it was majorly limited by the very small sample sizes of other ethnic groups within UK Biobank participants.

Overall, we recognize the lack of ethnicity diversity is a potential limitation and have discussed it accordingly in our manuscript. Please refer to page 9, line 334-336, where we elaborate on this point.

Reviewer #2 (Remarks on code availability):

The code is not there, there is just a code for vaccine effectiveness. Probably will be there on publication.

Response:

Thanks, we will make our codes public once accepted.

Reviewer #3 (Remarks to the Author):

The authors have addressed my previous concerns except for the interpretation of Fig 1A from Cromer et al., 2023. In figure 1A provided in the rebuttal by the authors, the point at which the curves touch the y-axis is well below the level of detection of neutralising antibodies in most assays- hence why there are no data points this low on the curve, the modelling extrapolated the curves to this point. As in my previous comment, the lowest reported/detectable levels of neutralising antibodies are ~20% of convalescent titres, or somewhere between 0.125 and 0.25 on the x-axis of Fig 1a, corresponding to protection efficacies of 50% and 80% for symptomatic and severe disease respectively.

The discussion should be amended to reflect this.

Response:

We sincerely thank the reviewer for their insightful comment. In response, we have made the correction in our manuscript, provided below and page 9 in the manuscript.

*Interestingly, a recent meta-analysis of 15 studies modelled that efficacy of detectable neutralizing antibodies may be approximately **50%** against infection and **80%** against severe COVID-19. The numerical figures of our estimated antibody protection are lower than the prior extrapolated ones, but it is critical to acknowledge key differences between studies. Firstly, neutralizing antibodies constitute only a part of the overall antibody repertoire. Secondly, the COVID-19 vaccines' efficacy and effectiveness may numerically differ in nature. **Thirdly, the scale of mendelian randomization estimates for exposure represents the odds ratio per 1 unit increase in the log odds of genetic liability to seroconversion, which is distinct from the nominal seropositivity (yes versus no).***

REVIEWER COMMENTS

Reviewer #1 (Remarks to the Author):

I thank the authors for providing a detailed explanation to the comments forwarded previously. Nonetheless, there are few items that they need to address:

1) The use of Fortress LFT cannot be justified for phenotyping cohort participants as antibody producers in response to SARS-CoV-2 vaccination given the fact that Fortress LFT is not able to distinguish between infection- and vaccination-induced antibody response.

2) Basically, the antibody immune response to SARS-CoV-2 is T cell-dependent whereby T cells become activated through the binding of TCRs of CD4+Th cells to peptide-MHCII complexes of APCs. The investigators insist that there is an association between MHC class I and antibody response, directly or indirectly. They should then provide justification supporting their hypothesis with an appropriate citation(s) to cement their argument. The reviewer agrees with the notion that there is an association between MHC-I and vaccine induced immunogenicity that could be the result of MHC-I:CD8TCR interactions, but independent of MHC-II:CD4TCR complex interaction, as has been rightly commented in the initial review and addressed by the authors as well in their revised version #1 of the manuscript.

3) While this manuscript is under review, a recent report by Bian C et. al. (Am J Hum Gen 2024;111:181), based on the same UKBB cohort, demonstrated an association between antibody production and HLA-II variants. In light of this recent findings conducted in the same cohort, but different results, the investigators of the current research work need to discuss the similarities & differences between the two reports.

Reviewer #2 (Remarks to the Author):

Ancestry (genetic) and ethnicity (cultural) are not the same thing and should not be confused. From one I can't infer the other, therefore "caucasian ethnicity confirmed by genotype" it is conceptually wrong because I can't infer your ethnicity based on your genotype and should be changed by "European ancestry confirmed by genotype".

I appreciate that in UK Biobank there isn't a lot of data from non-European ancestries, but the authors have added data from them in their analysis, which could alter the results if the effects were ancestry-dependant. Instead of doing a meta-analysis, the authors argue that the effect is the same in all ancestries and it does not affect the results. It is a very easy analysis to calculate the effects in non-European ancestries in HLA genes and show that they do not differ (and I appreciate that the error will be big). Otherwise, if other ancestries are not big enough to have an effect, why use them? Why not use EUR only individuals?

It may be the case that there are not ancestry-specific differences in the phenotype exactly because there are ancestry differences in effect. e.g. if the effect allele is much more common in one ancestry but the effect is smaller, then it may not be seen as differences in the response.

Reviewer #2 (Remarks on code availability):

Code available on publication

REVIEWER COMMENTS

Reviewer #1 (Remarks to the Author):

I thank the authors for providing a detailed explanation to the comments forwarded previously. Nonetheless, there are few items that they need to address:

1) The use of Fortress LFT cannot be justified for phenotyping cohort participants as antibody producers in response to SARS-CoV-2 vaccination given the fact that Fortress LFT is not able to distinguish between infection- and vaccination-induced antibody response.

Response: Thank you for your comment. While the test itself doesn't differentiate between an antibody response due to a COVID-19 infection or vaccination, we can confidently infer the cause. Our study participants included merely those without a history of SARS-CoV-2 infection. Thus, any detected SARS-CoV-2 antibodies should be attributed to vaccination. We've already elaborated on this key information in our manuscript. Please refer to page 24, lines 40, 42-46.

2) Basically, the antibody immune response to SARS-CoV-2 is T cell-dependent whereby T cells become activated through the binding of TCRs of CD4+Th cells to peptide-MHCII complexes of APCs. The investigators insist that there is an association between MHC class I and antibody response, directly or indirectly. They should then provide justification supporting their hypothesis with an appropriate citation(s) to cement their argument. The reviewer agrees with the notion that there is an association between MHC-I and vaccine induced immunogenicity that could be the result of MHC-I:CD8TCR interactions, but independent of MHC-II:CD4TCR complex interaction, as has been rightly commented in the initial review and addressed by the authors as well in their revised version #1 of the manuscript.

Response: The comment has been previously addressed. No action is needed in current version.

3) While this manuscript is under review, a recent report by Bian C et. al. (Am J Hum Gen 2024;111:181), based on the same UKBB cohort, demonstrated an association between antibody production and HLA-II variants. In light of this recent findings conducted in the same cohort, but different results, the investigators of the current research work need to discuss the similarities & differences between the two reports.

Response: Thank you for updating us on the most recently published research that used the same source population as ours. We have discussed their findings in our manuscript as below:

Most recently, in a study that used the same source data as ours, the authors reported four independent alleles: DRB1*13:02, DQA1*01:01, DPB1*04:01, and DQB1*02:01, associated with the antibody response to the initial dose. All these significant alleles, except for DPB1*04:01, were identified in present study, despite notable differences in participant eligibility criteria, cohort definition, antibody positivity determination windows, and analytic methods. Nevertheless, our analysis

pinpoints DQB1*06:04 and DRB3*01:01 as the most likely causal HLA alleles, which are distinct from but in high linkage disequilibrium with DRB1*13:02 and DQB1*02:01, respectively, as found in their study. Further functional experiments are needed to corroborate or refute these statistical fine mapping findings.

Reviewer #2 (Remarks to the Author):

Ancestry (genetic) and ethnicity (cultural) are not the same thing and should not be confused. From one I can't infer the other, therefore "caucasian ethnicity confirmed by genotype" it is conceptually wrong because I can't infer your ethnicity based on your genotype and should be changed by "European ancestry confirmed by genotype".

I appreciate that in UK Biobank there isn't a lot of data from non-European ancestries, but the authors have added data from them in their analysis, which could alter the results if the effects were ancestry-dependant. Instead of doing a meta-analysis, the authors argue that the effect is the same in all ancestries and it does not affect the results. It is a very easy analysis to calculate the effects in non-European ancestries in HLA genes and show that they do not differ (and I appreciate that the error will be big). Otherwise, if other ancestries are not big enough to have an effect, why use them? Why not use EUR only individuals?

It may be the case that there are not ancestry-specific differences in the phenotype exactly because there are ancestry differences in effect. e.g. if the effect allele is much more common in one ancestry but the effect is smaller, then it may not be seen as differences in the response.

Response: Thanks for the insightful comment.

We have modified below texts in the manuscript:

*Our analysis was restricted to participants to those of the **Caucasian ancestry as confirmed by genotype** (individuals listed in UK Biobank data filed 22006).*

Please refer to page 28, lines 140-141

Also, following your suggestion, we conducted a sensitivity analysis to examine potential heterogeneity of HLA effects. The results presented below are reassuring and show that the effects are likely ancestry-independent. We have included these

results in the manuscript. Please refer to page 6, lines 183-184. Page 28, lines 155-156.